# Exploiting human fucosyltransferase 8 allostery with a covalent inhibitor for core fucosylation suppression

Jiheng Jiang [1,2,3], Dongyang He[2,3], Mengyu Ke[2,3], Jinhua Qin[1,2], Guang Yang[1,2], Biao Yu [1,2] ✉, Jing Wang [1,2] ✉ & Pengfei Fang [1,2] ✉

Core fucosylation, catalyzed by fucosyltransferase 8 (FUT8), plays critical roles in cancer progression, immune evasion, and drug resistance, making it a compelling therapeutic target. However, development of selective FUT8 inhibitors has been hindered by shared substrate specificity of fucosyl-transferases. Here, we report the discovery of a previously unrecognized allosteric site on FUT8 and the development of a low-toxicity covalent inhibitor, CAIF (stearic acid-N-hydroxysuccinimide ester-dimethylimidazolium bromide), through structure-based drug design. High-throughput screening and crystallographic studies reveal that small molecules such as NH125 bind to a channel-like allosteric pocket, inducing conformational changes that disrupt FUT8 activity. Leveraging these insights, we design CAIF to covalently target lysine K216 within the allosteric site. CAIF exhibits minimal cytotoxicity and significantly inhibits core fucosylation and cancer cell invasion in cellular assays. This work establishes CAIF as a lead compound for further optimization and development, offering a framework for targeting glycosyltransferases through allosteric and covalent inhibition strategies.

Glycosylation, a common post-translational modification of proteins, plays pivotal roles in regulating diverse physiological functions in living organisms[1]. Core fucosylation, specifically catalyzed by fucosyl-transferase 8 (FUT8) in mammals[2], involves the transfer of fucose from guanosine 5′-diphospho-β-ʟ-fucose (GDP-Fuc) to the innermost N-acetylglucosamine residue of N-glycans attached to asparagine residues in proteins[3–5]. Dysregulation of FUT8 expression is linked to the progression of several diseases, notably including metastatic melanoma[6], triple-negative breast cancer[7], head and neck squamous cell carcinoma[8], and non-small cell lung cancer[9]. In these conditions, elevated FUT8 levels result in increased core fucosylation of glycoproteins such as L1 cell adhesion molecule, B7 homolog 3, and sema-phorin 7 A, which promotes tumor invasion, metastasis, and immune

evasion[6–9]. These findings underscore the clinical significance of targeting FUT8 activity.

Inhibition of FUT8, through genetic knockout or small-molecule inhibitors, enhances anti-tumor immunity by downregulating immune checkpoint molecules and reducing tumor growth and metastasis. For example, blocking core fucosylation of programmed cell death protein 1 (PD-1) by inhibiting FUT8 reduces cell-surface expression of PD-1 and enhances T cell activation, leading to more efficient tumor eradication[10]. FUT8 is thus considered a promising therapeutic target for cancer immunotherapy[11]. Additionally, FUT8-mediated core fuco-sylation plays a critical role in promoting targeted drug resistance and chemoresistance[12,13], as its expression is broadly correlated with resistance to multiple targeted therapies, especially epidermal growth

[1]Key Laboratory of Glyco-drug Research of Zhejiang Province, School of Chemistry and Materials Science, Hangzhou Institute for Advanced Study, University of Chinese Academy of Sciences, Hangzhou, China. [2]State Key Laboratory of Chemical Biology, Shanghai Institute of Organic Chemistry, University of Chinese Academy of Sciences, Chinese Academy of Sciences, Shanghai, China. [3]These authors contributed equally: Jiheng Jiang, Dongyang He, Mengyu Ke. ✉e-mail: byu@sioc.ac.cn; jwang@sioc.ac.cn; fangpengfei@sioc.ac.cn

factor receptor (EGFR) and insulin-like growth factor receptor inhibitors[12]. Suppression of FUT8 through RNA interference enhances drug-induced cell killing and sensitizes resistant clones, while its overexpression rescues resistance, highlighting its pivotal role in modulating drug sensitivity[12]. Beyond its role in immune regulation and drug resistance, core fucosylation also plays a critical role in regulating tumor necrosis factor (TNF)-related apoptosis inducing ligand (TRAIL)-induced apoptosis. FUT8 knockdown enhances sensitivity to TRAIL in SW480 cells and potentiates mitochondrial-dependent apoptosis via caspase-9 activation, highlighting its role in modulating cell death pathways in colorectal cancer[13]. Furthermore, FUT8-mediated core fucosylation enhances RNA viral replication by suppressing retinoic acid-inducible gene I (RIG-I)-mediated antiviral immunity and activating EGFR signaling[14], highlighting its significance as a potential target in antiviral therapy. Core fucosylation can reduce the efficacy of antibody-dependent cellular cytotoxicity[15–17], necessitating the suppression or inhibition of FUT8 in the production of therapeutic antibodies and cellular immunotherapies[16,18].

To rationally develop effective inhibitors targeting FUT8, a deep understanding of its structure and catalytic mechanism is essential. FUT8 is a type II transmembrane protein composed of an N-terminal cytoplasmic region, a transmembrane domain, a stem region (V30–L108), a coiled-coil domain (G109–T173), a GT-B catalytic domain (D174–N503 and E563–K575), and an Src homology 3 (SH3) domain (N503-E563)[19–21]. FUT8 forms a dimer in solution, with the dimerization interface comprising an extended four-helix bundle contributed by two helices from each monomer, further stabilized by interactions between the bundle and the SH3 domain[22,23]. The catalysis by FUT8 resembles an SN2-type displacement mechanism, involving a series of loops and α-helices that form the binding sites[22]. A pocket formed by one of these loops and the SH3 domain is responsible for recognizing the branched glycan and establishing specific contacts with the α−1,3-GlcNAc branch, which is essential for catalysis[23]. The interaction between GDP-Fuc and the donor-binding site primarily involves hydrogen bonds between the nucleotide base and the donor-binding pocket. Upon GDP-Fuc binding, the highly flexible loop region of FUT8 becomes ordered, and an R365−D368−R441 motif encloses the sugar nucleotide, locking the donor in place[23,24]. Binding of the glycan acceptor via the O6 hydroxyl group of the GlcNAc residue, which acts as the nucleophile, leads to deprotonation of E373. A subsequent SN2-type attack on the C1 carbon of the donor sugar results in fucose transfer, with proton shuttling from E373 via K369 to the β-phosphate leaving group, thereby completing the glycosylation reaction[23,24]. These structural insights reveal distinct binding sites for both the donor and acceptor substrates, providing a critical framework for designing specific inhibitors.

Over the past three decades, considerable effort has been directed towards the development of FUT8 inhibitors[25–27], encompassing both substrate analogs and non-sugar-related compounds. As the fucosyltransferase family uses GDP-Fuc as a shared donor substrate[25], fucose and GDP-Fuc analogs, such as 2-fluorofucose (2FF)[28], β-carbafucose[18], and fluorinated rhamnosides[29], act as pan-inhibitors of fucosylation. However, these pan-inhibitors may cause significant side effects, as seen with the discontinuation of 2FF after thromboembolic events were reported during phase I clinical trials for solid tumors[28].

Previous studies have also made significant strides in identifying non-sugar FUT8 inhibitors, such as FDW028, through virtual screening of FUT8's active site, demonstrating low micromolar binding affinity[30]. In a recent study, high-throughput screening and structural optimization led to the discovery of a FUT8-selective inhibitor that exhibits strong binding to FUT8 in the presence of GDP[31]. Despite these promising developments, the exact binding sites of these inhibitors on FUT8 remain unknown, and their mechanisms of action (MoAs) have not been clearly defined. Together, these studies underscore the progress made in FUT8 inhibitor discovery while highlighting the

critical need for further mechanistic insights to fully realize their therapeutic potential.

Given the important value of FUT8 inhibitors and the current lack of inhibitors with well-defined MoAs, our study aims to address this critical gap by integrating advanced screening techniques with structural biology approaches. Through high-throughput screening of FUT8 enzymatic activity, X-ray crystallography, and chemical synthesis, we have discovered a series of FUT8 inhibitors. Notably, we have identified an allosteric covalent regulatory site on FUT8, which, to our knowledge, has not been previously reported in human fucosyltransferases. This breakthrough provides valuable insights into the design of highly selective inhibitors within the fucosyltransferase family, offering a strategic framework for future therapeutic development.

## Results

### Identification and characterization of FUT8 inhibitors

To discover core fucosylation inhibitors, we expressed and purified the human FUT8 (68-575) (Supplementary Fig. 1a–c) and established robust luminescence-based activity assay for high-throughput screening (HTS) (Supplementary Fig. 1d,e). Using this assay, we performed an HTS of approximately 6248 compounds from Selleck Chemicals (Fig. 1a and Supplementary Fig. 1f). The screening demonstrated excellent quality, with average Z′-factors of 0.85 and 0.90 for the two independent screening sets (Fig. 1a,b and Supplementary Fig. 1f). From the initial hits, we identified three compounds—mitoxantrone, NH125, and chlorhexidine (CHX)—that exhibited consistent inhibitory activity in both the primary luminescence assay and a secondary validation assay (Transcreener GDP FI assay) (Supplementary Table 1 and Fig. 1c). Dose-response experiments revealed that NH125 and CHX inhibited FUT8 with $IC_{50}$ values of about 15.3 and 14.4 μM, respectively, while mitoxantrone was less potent (Fig. 1d, Supplementary Fig. 2, and Supplementary Table 2). As mitoxantrone is a frequent hit in screenings against diverse targets, suggesting potential non-specificity, we focused subsequent studies on NH125 and CHX.

We next evaluated the cytotoxicity and the ability to inhibit cellular core fucosylation of NH125 and CHX on A375 cells. NH125 and CHX showed some toxicity to inhibit cell proliferation in A375 cells, with $IC_{50}$ values of about 13.2 and 6.9 μM, respectively (Supplementary Fig. 3a). Using fluorescein isothiocyanate (FITC) labeled lentil lectin (LCA), which specifically recognizes core fucosylated N-glycans, we found that NH125 and CHX both significantly alleviated the core fucosylation on A375 cell surface in confocal microscopy and flow cytometry experiments (Fig. 1e and Supplementary Fig. 3b,c). Consistent with previous discovery that inhibiting or knocking out FUT8 suppresses the metastatic and invasive capabilities of melanoma[6], both CHX and NH125 demonstrated robust inhibitory effects on the in vitro invasion of A375 cells (Fig. 1f and Supplementary Fig. 3d,e). These phenotypes, potentially arising from the dual effects of suppressed glycosylation and inhibited cell growth, suggest that NH125 and CHX represent promising lead compounds for the development of FUT8-targeting therapeutics. Further optimization of these compounds to minimize their cell growth inhibitory effects while retaining FUT8-targeting activity will help more specifically elucidate the functional roles of core fucosylation.

### NH125 and CHX bind to an allosteric inhibitory site of FUT8

To investigate the inhibitory mechanisms of NH125 and CHX on FUT8, we determined the crystal structures of the complexes formed by these two small molecules with FUT8. The FUT8-CHX complex was solved to a resolution of 3.0 Å, and the FUT8-NH125 complex was solved to a resolution of 2.5 Å (Supplementary Table 3). They both belong to the C121 space group with similar cell parameters. The crystal structures revealed a homodimer organization facilitated by the first two N-terminal α-helices (namely, the stem region) of FUT8

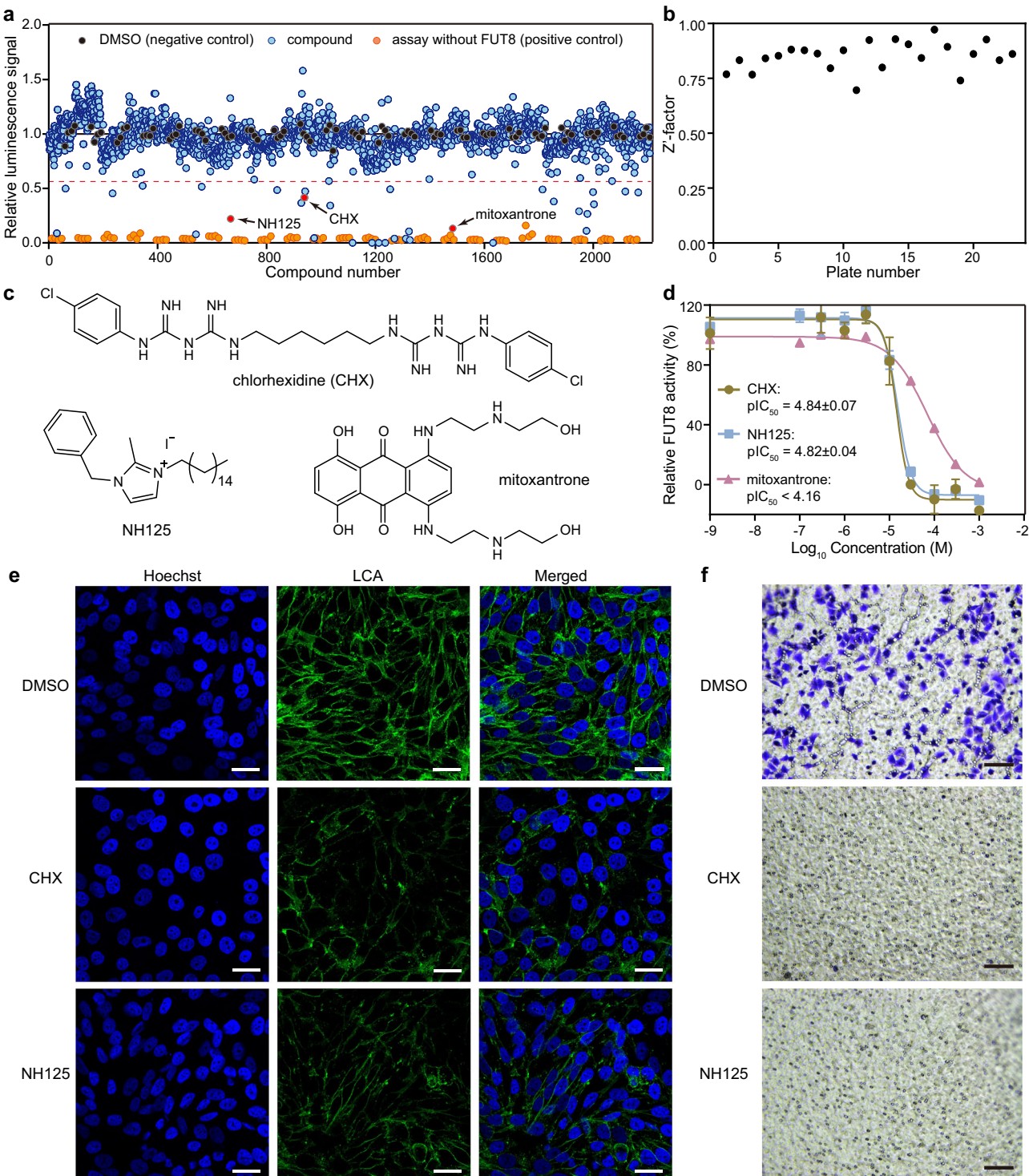

**Fig. 1 | Identification and characterization of Fut8 inhibitors. a** Results of the primary screening assay of approximately 2300 compounds from the FDA-approved Drug Library. Data are shown for the tested compounds (concentration: 100 μM, blue dots), positive control (experimental system without FUT8, orange dots), negative control (DMSO, black dots). Compounds demonstrating normalized values < 0.6 (dashed line) were selected for further validation, and red dots indicate validated potential hits. **b** Scatter plot of the Z'-factor, calculated from the HTS data shown in panel **a**, which evaluates the quality of the HTS assay in the absence of any test compounds. The average Z'-factor was 0.85. **c** The chemical structures of chlorhexidine (CHX), NH125 and mitoxantrone. **d** Concentration-response curves showing the inhibition of FUT8 activity by the identified hits. Corresponding IC₅₀ values of CHX, NH125, and mitoxantrone are 14.4 μM, 15.3 μM,

and > 69.5 μM, Data points show the mean ± SD of quadruplicate technical repeats from one representative independent experiment. The dose-response curve from another independent replicate is provided in Supplementary Fig. 2. **e** Representative confocal microscopy images of A375 cells treated with DMSO (control), CHX (10 μM), or NH125 (5 μM) for 72 hours. Cells were stained with FITC-conjugated of LCA (green) to visualize fucosylation and Hoechst (blue) to label nuclei. Scale bar: 25 μm. Images shown are from one of two independent experiments that yielded similar results. **f**. Migration and invasion ability of A375 cells assessed by the in vitro invasion assay after treatment with DMSO (control), CHX (10 μM), or NH125 (5 μM). Scale bar: 100 μm. Images shown are from one of two independent experiments that yielded similar results. Source data are provided as a Source data file.

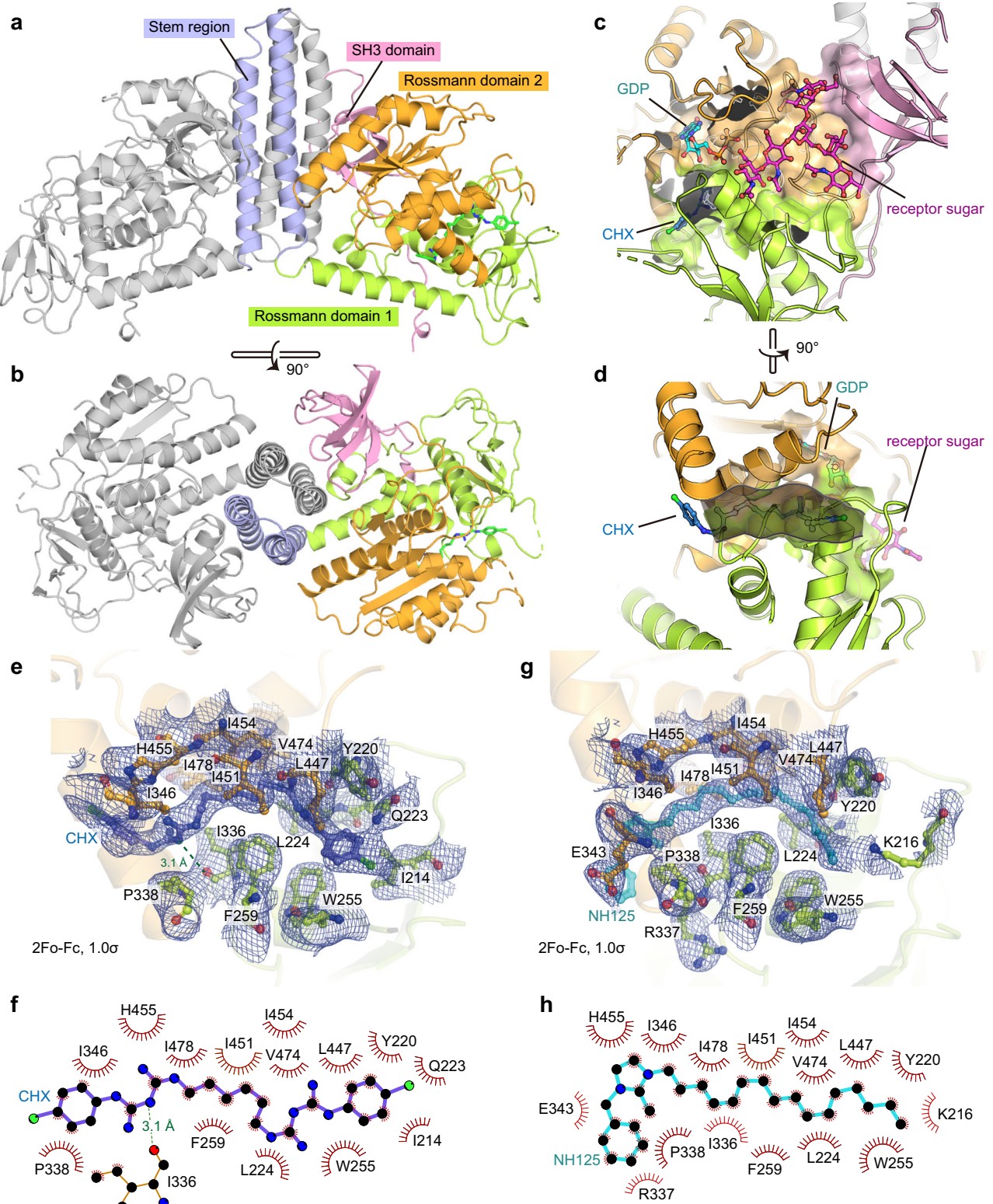

(Fig. 2a,b). These helices arrange to form a 4-helix bundle, which aligns with previously reported structures of FUT8[21–23]. The active center of FUT8 is situated at the intersection of Rossmann 1, 2, and SH3 domains[22,23]. Specifically, the binding site of GDP-Fuc is mainly formed by the Rossmann domain 2, while the receptor sugar binds to the furrow between the three domains of Rossmann 1, 2 and SH3 (Fig. 2c,d).

Surprisingly, neither NH125 nor CHX binds to the substrate binding site of FUT8. Instead, they induce a channel-like binding pocket between the Rossmann domains 1 and 2 of FUT8, located in proximity to the GDP-Fuc binding site (Fig. 2c,d). Both NH125 and CHX possess a chain-like structure and are bound by FUT8 almost exclusively through hydrophobic interactions. The residues directly involved in binding include I214, K216, Y220, Q223, L224, W255, F259, I336, R337, P338, E343, I346, L447, I451, I454, H455, V474, I478, etc. (Fig. 2e–h). Notably, the only polar interaction is a hydrogen bond between the I336 main chain oxygen atom and an imino nitrogen atom in CHX (Fig. 2e,f).

**Fig. 2 | NH125 and CHX target an allosteric inhibitory site of FUT8. a–b** Overview of the structure of FUT8 in complex with CHX (green sticks). Chain A of FUT8 is shown as a gray cartoon (left). In Chain B (right), the stem region is depicted in light blue, Rossmann domain 1 in lemon yellow, Rossmann domain 2 in orange, and the SH3 domain in pink. **c** The substrate binding sites of FUT8 are shown as a transparent surface. GDP and the receptor glycan, modeled using PDB: 6TKV as a reference, are depicted as cyan and purple sticks, respectively. **d** The binding pocket of CHX is shown as a shaded transparent surface. The CHX molecule is depicted as blue sticks. **e** Zoomed-in view of the FUT8-CHX interactions. Residues in Rossmann domain 1 interacting with CHX are shown as lemon sticks, while those in Rossmann domain 2 are shown as orange sticks. CHX is shown as blue sticks. A

hydrogen bond is represented as a green dashed line. The 2Fo-Fc electron density (blue meshes contoured at 1.0 σ) is shown together with the structure model. **f** Two-dimensional representation of the FUT8-CHX interaction, generated using LigPlot[56]. A hydrogen bond (3.1 Å) is formed between CHX and I336, while other interactions are primarily hydrophobic. **g** Zoomed-in view of the FUT8-NH125 interactions. Residues in Rossmann domain 1 interacting with NH125 are shown as lemon sticks, while those in Rossmann domain 2 are shown as orange sticks. NH125 is shown as cyan sticks. The 2Fo-Fc electron density (blue meshes contoured at 1.0 σ) is shown together with the structure model. **h** Two-dimensional representation of the FUT8-NH125 interaction, generated using LigPlot[56]. Almost all the interactions are primarily hydrophobic.

Importantly, the binding of CHX or NH125 would not result in direct steric rejection of the FUT8 substrate GDP-Fuc or the receptor sugar from the structure. Therefore, they are identified as allosteric inhibitors of FUT8.

## Allosteric site binders change the conformation and dynamics of FUT8

In order to further understand the MoAs of CHX and NH125 as allosteric inhibitors, we compared the FUT8-CHX and FUT8-NH125 complexes with the FUT8-GDP-sugar complex (PDB code: 6TKV, referred to as *holo*) reported previously[23].

In the *holo* structure, the $_{247}$NWRYATG$_{253}$ loop plays a role in binding the sugar donor substrate. Specifically, the phenol hydroxyl group on the Y250 side chain and the two oxygen atoms on GDP ribose can form hydrogen bond interactions in the *holo* structure. However, in both the FUT8-CHX and FUT8-NH125 structures, the inhibitors induced significant conformational changes in the Y220 residue. If the $_{247}$NWRYATG$_{253}$ loop was structured as in the *holo* state in the inhibitor-bound structures, the side chain rotamer of Y220 would clash with Y250 (Fig. 3a, and Supplementary Fig. 4a,b). Furthermore, CHX or NH125 would directly clash with W248 under the same condition (Fig. 3a, and Supplementary Fig. 4a,b). Consequently, in the inhibitor-bound structures, the $_{247}$NWRYATG$_{253}$ loop becomes disordered, thereby losing its interactions with the sugar donor substrate.

It is worth mentioning that in the FUT8-CHX structure, only one molecule of the FUT8 dimer binds to a CHX, while the other molecule remains in its *apo* state (Supplementary Fig. 5a,b). In the FUT8-NH125 structure, both molecules (chains A and B) of the dimer bind to NH125 (Supplementary Fig. 5c,d). Overall, the architecture of the FUT8 protein remains largely unchanged, with no major domain-scale conformational differences observed between the FUT8-NH125 and FUT8-*holo* complex structures (Fig. 3b). Consistent with the location of the inhibitor-binding pocket distal to the dimer interface, ligand binding also induces no notable structural perturbations at the FUT8 dimer interface (Fig. 3b). Despite this overall structural stability, a detailed comparison of the FUT8_*apo*, FUT8-CHX, FUT8-NH125_A, FUT8-NH125_B, and FUT8-*holo* structures reveals intrinsic flexibility in the $_{215}$NKG$_{217}$ loop (Fig. 3c). Specifically, in the FUT8-CHX and FUT8-NH125_B structures, K216 undergoes a conformational change, flipping from forming hydrogen bonds with D295 and the N-acetylglucosamine moiety of the sugar receptor, to having its side chain directed towards the allosteric pocket (Fig. 3c). This change can potentially be utilized in future drug design, as the primary amino group in the side chain of lysine exhibits nucleophilic properties and can be specifically targeted by FUT8 allosteric site binders that carry corresponding covalent warheads.

It has been demonstrated that protein kinetics influences the catalytic efficiency and drug resistance of enzymes[32,33]. Using the FUT8-NH125 structure as the initial conformation, and molecular dynamics (MD) simulations were performed both in the presence and absence of bound NH125 (Supplementary Fig. 6a,b). The per-residue Root Mean Square Fluctuation (RMSF) profile from our MD simulations and the

B-factor profile from our crystal structure show a remarkably high degree of concordance (Supplementary Fig. 6c). This strong correlation validates the reliability of our MD simulations and confirms that the dynamic regions captured in the crystal lattice are consistent with the solution dynamics we observed in silico. The RMSF analysis revealed that NH125 binding induces asymmetric changes in flexibility across the structure. Although the overall RMSF increased in the NH125-bound state—likely due to the intrinsic stochasticity of molecular dynamics simulations—this increase did not exceed the standard deviation observed across three fully independent replicates, indicating that the changes are not statistically significant. Notably, two loops critical for natural substrate binding, $_{366}$RTDKVGTEA$_{374}$ and $_{431}$SISWSAGLHNRYTENS$_{446}$, exhibited the most pronounced increases in flexibility (Supplementary Fig. 6d). In contrast, a slight decrease in RMSF was observed in some regions, such as parts of the SH3 domain that are involved in receptor glycan binding (Supplementary Fig. 6d). Furthermore, the $_{215}$NKG$_{217}$ loop exhibited higher flexibility than adjacent residues, but its RMSF showed no significant change upon NH125 binding (Supplementary Fig. 6d). These results indicate that the dynamic nature of the $_{215}$NKG$_{217}$ loop is an innate property of the FUT8 structure and is not induced or significantly altered by allosteric inhibitor binding. This suggests that the functional role of this loop may be related to its intrinsic conformational plasticity, which appears to be preserved rather than perturbed upon NH125 binding. These dynamic changes provide a potential mechanistic explanation for the allosteric inhibition of substrate binding. Importantly, residues at the extensive dimer interface showed minimal changes in flexibility upon NH125 binding (Supplementary Fig. 6d), suggesting that the allosteric inhibition mechanism operates independently of quaternary structural perturbations.

In addition, comparison of the free energy landscape (FEL) projected along the first two principal components (PC1 and PC2) revealed that NH125 binding remodels the conformational ensemble of FUT8 (Fig. 3d and Supplementary Fig. 7). In the Apo state, the FEL exhibits four metastable basins with their relative free energies of 0 kJ/mol (upper-right), 0.63 kJ/mol (upper-middle), 0.42 kJ/mol (central), and 9.15 kJ/mol (lower-middle). These basins reflect a dynamic conformational distribution, with several low-energy states likely representing functionally relevant conformations accessible during catalytic cycles. The basin with higher free energy (~9.15 kJ/mol) may correspond to a less populated, transient state. Upon NH125 binding, the FEL is significantly reconfigured, also featuring four distinct basins located with relative minima at 0.95 kJ/mol (upper-left), 0 kJ/mol (upper-middle), 9.78 kJ/mol (central), and 1.42 kJ/mol (lower-middle) (Fig. 3d). Notably, NH125 stabilizes a different global minimum in the upper-middle region (0 kJ/mol)—a state not prominently sampled in the Apo form—while destabilizing several native basins, as reflected by increased energies in other regions. Although conformational diversity is retained, the reshaped energy landscape features altered barrier heights and basin distributions. This remodeling suggests that NH125 binding perturbs the thermodynamic preferences of FUT8, potentially trapping the enzyme in non-productive states and disrupting the

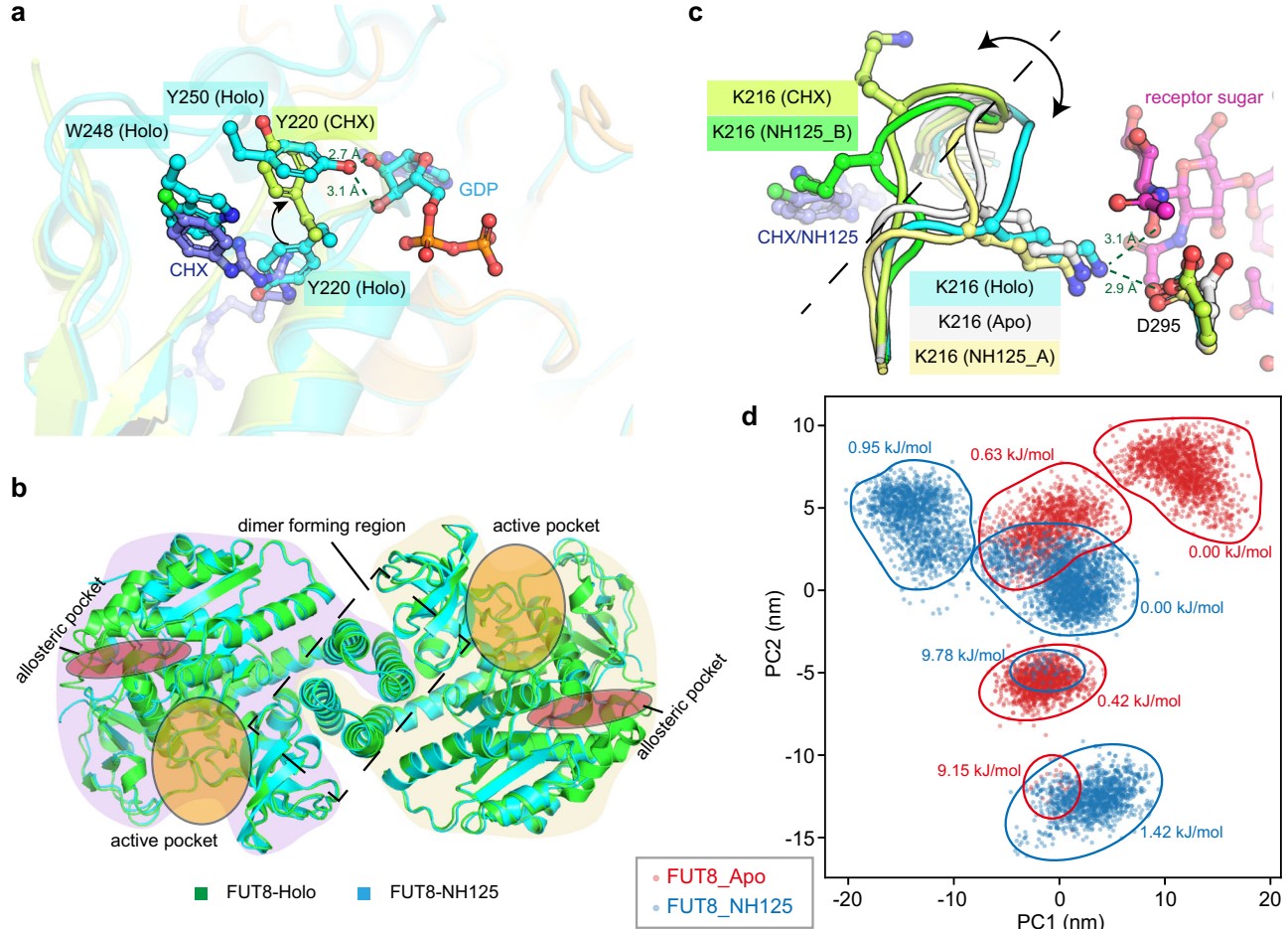

**Fig. 3 | Allosteric site binders alter the conformation and dynamics of FUT8.**
**a** The FUT8-CHX structure (this study) is superimposed on the FUT8 *holo* Structure (PDB code: 6TKV). CHX binding induces conformational changes in the $_{247}$NWRYATG$_{253}$ loop. CHX, GDP, and key residues are shown as sticks. The main structures of the protein are represented as cartoons. **b** The FUT8-NH125 (this study, blue) and FUT8 *holo* (PDB code: 6TKV, green) structures are superimposed. The overall structures align closely, with no major conformational changes observed in the global fold upon NH125 binding. The allosteric inhibitor binding site (indicated by a red oval) is located distally from the extensive dimer interface (indicated by a dashed box), indicating that the inhibition mechanism is independent of quaternary structural perturbations. **c** The FUT8 *apo*, FUT8-CHX, FUT8-NH125, and FUT8 *holo* structures are superimposed, highlighting the flexibility of

the $_{215}$NKG$_{217}$ loop. The receptor glycan and K216 are shown as sticks. The main structures of the protein are represented as cartoons. **d** Comparative conformational distribution of the Apo (red scatter plot) and NH125-bound (blue scatter plot) states, projected onto the first two principal components (PC1 and PC2) from a combined analysis of all six molecular dynamics trajectories aligned to a common reference. Encircled clusters represent metastable conformational states. The relative free energy value (in kJ/mol) for each cluster is annotated, with the global minimum for each system designated as 0 kJ/mol. NH125 binding dramatically reorganizes the conformational ensemble, stabilizing a different global minimum state and altering the stability of functional basins, effectively trapping the enzyme in a non-productive conformation and illustrating the structural basis for its allosteric inhibitory mechanism. Source data are provided as a Source data file.

conformational cycling required for catalysis. Therefore, our simulations indicate that NH125 may act as an allosteric inhibitor not only through binding but also by reprogramming the free energy landscape of FUT8, thereby impairing its functional dynamics.

### The allosteric site of FUT8 can be covalently targeted
The identification of allosteric sites offers a distinct strategy to selectively inhibit FUT8, potentially addressing the challenges posed by the shared substrate specificity among fucosyltransferases in the development of selective inhibitors. Since the allosteric regulatory site possesses a long, channel-like shape, we selected and tested four additional compounds with chain-like structures: olanexidine, sulfosuccinimidyl oleate sodium (SSO), crocetin, and crocin II (Fig. 4a). Among these four compounds, they demonstrated varying levels of inhibition against FUT8.

Notably, SSO completely suppressed the relative activity of FUT8, reducing it to baseline levels with an IC50 value of 36.8 μM (Fig. 4b, Supplementary Fig. 2d, and Supplementary Table 2). SSO is

distinguished by its unique structural composition, which includes an N-hydroxysuccinimidyl (NHS) ester moiety—a covalent warhead commonly used in molecular probes. This observation led us to hypothesize that SSO may also act as a covalent inhibitor of FUT8.

To elucidate the MoA of SSO, we conducted co-crystallization experiments involving FUT8 and SSO. The FUT8-SSO complex structure was solved to a resolution of 2.7 Å (Supplementary Table 4). In this structure, SSO occupies the allosteric sites, similar to CHX and NH125. And indeed, SSO functions as a covalent inhibitor of FUT8, covalently binding to the K216 residue of FUT8 (Fig. 4c,d and Supplementary Fig. 8), thereby leading to an irreversible inhibition of FUT8's activity. This finding is consistent with the prior functional observation that mutation of this residue (K216A) completely abrogates FUT8 catalysis, underscoring its essential role in the enzyme's function[23,24].

### Development of a low-toxic allosteric covalent inhibitor of FUT8
Given that both NH125 and SSO share a remarkably similar long straight-chain aliphatic structure, we subsequently employed a

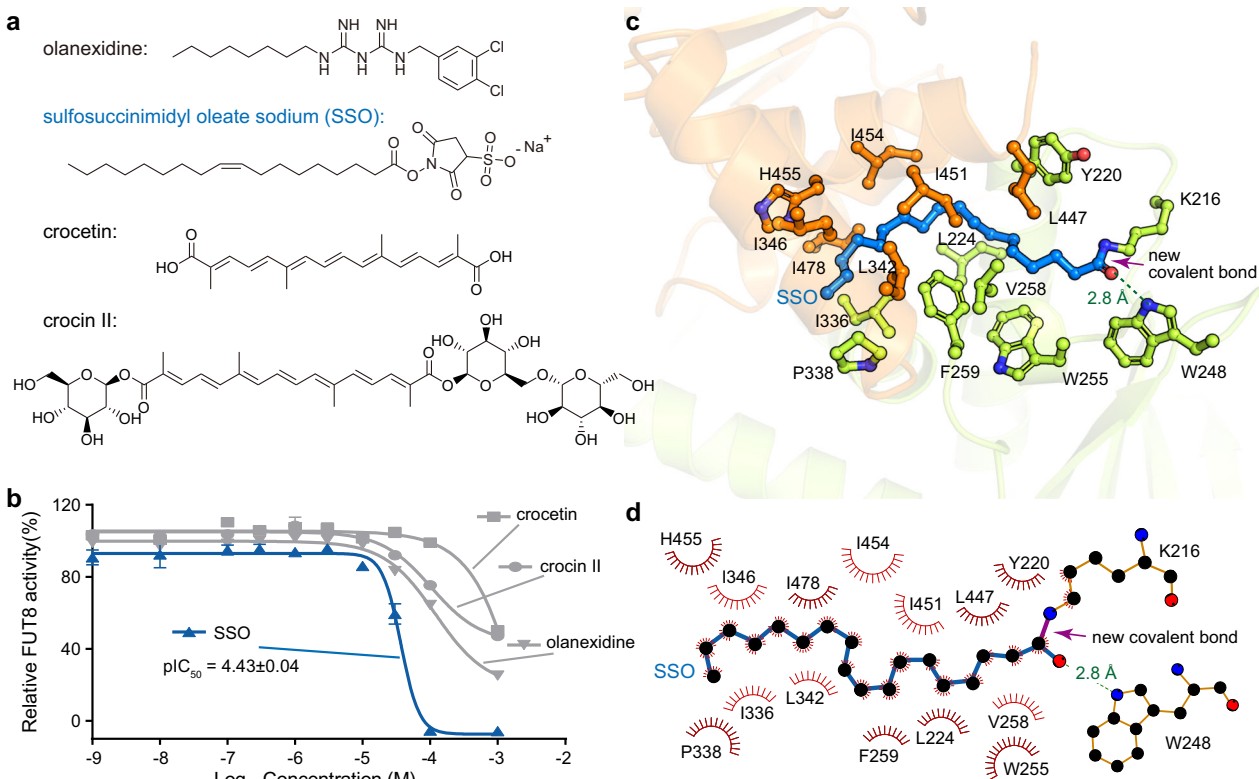

**Fig. 4 | Sulfosuccinimidyl oleate sodium (SSO) covalently binds to the allosteric site of FUT8. a** Chemical structures of chain-like compounds (olanexidine, SSO, crocetin, and crocin II). **b** Concentration-response curves showing the inhibition of FUT8 activity by chain-like compounds. Corresponding IC$_{50}$ values of SSO is 36.8 μM. Data points show the mean ± SD of quadruplicate technical repeats from one representative independent experiment. The dose-response curve from another independent replicate is provided in Supplementary Fig. 2d. **c** Zoomed-in view of the FUT8-SSO interactions. Residues in Rossmann domain 1 interacting with SSO are shown as lemon sticks, while those in Rossmann domain 2 are shown as orange sticks. SSO is shown as blue sticks. A hydrogen bond is represented as a green dashed line. **d** Two-dimensional representation of the FUT8-SSO interaction, generated using LigPlot[56]. SSO formed a covalent bond with K216. Source data are provided as a Source data file.

fragment merging strategy to develop an integrated compound that combines the advantageous features of both molecules for enhanced FUT8 inhibition (Fig. 5a). Structural analysis of FUT8-NH125 complexes (FUT8-NH125_A and FUT8-NH125_B) revealed conformational flexibility in the benzene moiety of NH125, suggesting its limited contribution to stable receptor interactions (Supplementary Fig. 8b). Consequently, we excluded this structural element in our compound design. Furthermore, to reduce potential nonspecific protein modifications, we removed the sulfonic acid group from the maleimide ring of SSO (Fig. 5a).

The resulting compound, 1-(18-((2,5-dioxopyrrolidin-1-yl)oxy)−18-oxooctadecyl)−2,3-dimethyl-1H-imidazol-3-ium bromide (Supplementary Note 1), demonstrated significantly improved FUT8 inhibitory activity, exhibiting an IC$_{50}$ value of 5.7 μM (Fig. 5b, Supplementary Fig. 2e,f, and Supplementary Table 2). This represents a 2.5-fold enhancement compared to NH125 (IC$_{50}$ about 15 μM) and a 6-fold improvement over SSO (IC$_{50}$ about 37 μM). To confirm that the observed inhibition was specifically due to targeting FUT8 and not the detection system, we performed additional control experiments. Incubation of varying concentrations of CAIF with the coupling enzymes and GDP substrate (in the absence of FUT8) revealed no inhibitory effect up to 1 mM (Supplementary Fig. 9a). We next evaluated the selectivity of CAIF against other members of the same protein family, FUT3 and FUT4, which represent the most likely potential sources of off-target activity. Despite this close phylogenetic relationship, the results are highly encouraging: CAIF exhibited no activity against FUT4 and only weak inhibition against FUT3 (Supplementary Fig. 9b,c). This translates to a remarkable selectivity index of over 114-

fold for FUT8 over FUT3. To further evaluate the selectivity of CAIF and assess its potential off-target effects, we computationally screened the human proteome for structural homologs of the GT-B domain of FUT8 using FoldSeek[34]. Only two proteins (Prob. > 0.23), POFUT1[35] and POFUT2[36], were identified as close structural homologs of FUT8. Despite low sequence identity (11% and 12%, respectively), their structural similarity suggested potential shared binding features. We subsequently expressed and purified both POFUT1 and POFUT2 and evaluated the inhibitory activity of CAIF. Remarkably, CAIF showed no significant inhibition against either POFUT1 or POFUT2, even at the highest tested concentration of 1 mM (Supplementary Fig. 9d,e). This finding robustly underscores the advantage of targeting the unique allosteric pocket of FUT8 for achieving selective inhibition.

To elucidate the MoAs of this putative covalent allosteric inhibitor of FUT8 (CAIF), we also determined the co-crystal structure of the FUT8-CAIF complex at 2.0 Å resolution (Supplementary Table 4). The structural data confirmed our design rationale, revealing CAIF binding at the allosteric site. The inhibitor occupies a pocket formed by Y220, L224, W248, W255, V258, F259, I336, P338, E343, I346, L447, I451, I454, H455, and I478, and forms a covalent amide bond with the ε-amino group of K216 (Fig. 5c,d and Supplementary Fig. 8c,d). To functionally validate the importance of Y220 suggested by the structure, we tested the inhibitory activity of CAIF against the Y220A mutant. Interestingly, its inhibitory activity was drastically reduced, with an IC$_{50}$ value exceeding 503 μM (Supplementary Fig. 9f). This represents a greater than 83-fold loss of potency compared to the wild-type protein (IC$_{50}$ about 6 μM). This is consistent with our proposed mechanism in which the binding of CAIF triggers a conformational change that positions

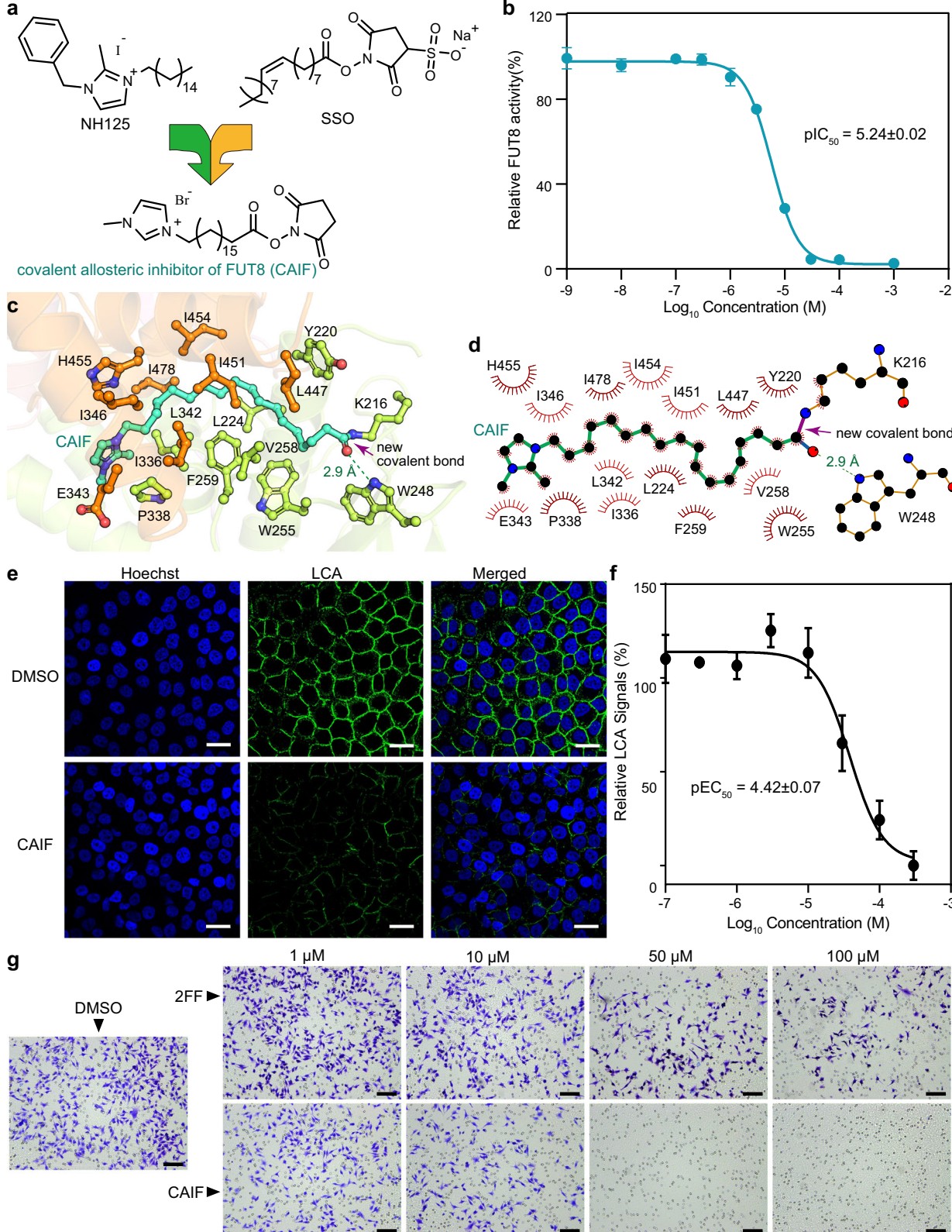

Y220 to sterically clash with Y250, thereby destabilizing the active state of FUT8.

We subsequently evaluated the cytotoxicity profile of CAIF across four distinct cell lines: HeLa, A375, GAK, and HEK293T. The cytotoxicity assays revealed that CAIF exhibited minimal cellular toxicity with all tested cell lines. Notably, even at the maximum tested concentration of 500 μM, CAIF showed no detectable cytotoxicity against HeLa

and GAK cells (Supplementary Fig. 10a). This favorable toxicity profile enabled us to investigate CAIF's effects on core fucosylation without interference from cell viability concerns. We employed high-content cellular imaging with fucose-specific lectins LCA and *Aleuria aurantia* lectin (AAL). CAIF treatment showed significant effect in inhibiting HeLa and A375 cells core fucosylation (Fig. 5e and Supplementary Fig. 10b). Quantitative analysis in GAK cells yielded an EC$_{50}$ of about

**Fig. 5 | Design and evaluation of the covalent allosteric inhibitor of FUT8 (CAIF). a** Design principle and structure of CAIF. **b** Concentration-response curves showing the inhibition of FUT8 activity by CAIF ($pIC_{50} = 5.24 \pm 0.02$, $IC_{50} = 5.7\,\mu M$). Data points show the mean ± SD of quadruplicate technical repeats from one representative independent experiment. The dose-response curve from two more independent replicate is provided in Supplementary Fig. 2. **c** Zoomed-in view of the FUT8-CAIF interactions. Residues in Rossmann domain 1 interacting with CAIF are shown as lemon sticks, while those in Rossmann domain 2 are shown as orange sticks. CAIF is shown as blue sticks. A hydrogen bond is represented as a green dashed line. **d** Two-dimensional representation of the FUT8-CAIF interaction, generated using LigPlot[56]. CAIF formed a covalent bond with K216. **e** Representative confocal microscopy images of HeLa cells treated with DMSO (control) or CAIF (50 μM) for 72 hours. Cells were stained with FITC-conjugated LCA (green) to visualize fucosylation and Hoechst (blue) to label nuclei. Scale bar: 25 μm. Images shown are from one of two independent experiments that yielded similar results. **f** Concentration-response curve of fucosylation levels in GAK cells treated with CAIF ($pEC_{50} = 4.42 \pm 0.07$, $EC_{50} = 38.4\,\mu M$). Image-based quantitative analysis was performed using the integrated fluorescence intensity of FITC-conjugated LCA. Data points show the mean ± SD of quadruplicate technical repeats. **g** Migration and invasion ability of HeLa cells assessed by the in vitro invasion assay after treatment with 2FF or CAIF (1 μM, 10 μM, 50 μM, 100 μM). Scale bar: 100 μm. Images shown are from one of two independent experiments that yielded similar results. Source data are provided as a Source data file.

40 μM (Fig. 5f and Supplementary Fig. 10c). In addition, CAIF treatment outperformed 2FF in suppressing HeLa cell migration and invasion in the in vitro invasion assay. (Fig. 5g and Supplementary Fig. 10d).

In summary, the development of CAIF represents a significant advancement in FUT8 inhibition, combining structural insights from NH125 and SSO to achieve enhanced potency, minimal cytotoxicity, and superior efficacy in modulating core fucosylation across multiple cancer cell lines. This inhibitor demonstrates robust activity in suppressing cancer cell migration and invasion, highlighting the broad therapeutic potential of FUT8 inhibition in diverse tumor models.

## Discussion

Glycosyltransferases, including FUT8, have proven challenging to target due to their structural complexity and shared substrate specificity, particularly the conserved sugar donor binding site[25,37]. This study overcomes these challenges by identifying a previously unrecognized allosteric site on FUT8, distinct from the substrate-binding pocket, and developing CAIF as a selective covalent inhibitor with a well-defined MoA. It is important to emphasize that the primary significance of this work lies in its role as a proof-of-concept study, which unveils a "druggable" allosteric site and suggests a covalent targeting strategy against FUT8. While CAIF serves as a tool compound to validate this mechanism, our current focus is on the target and mechanistic discovery rather than the immediate drug potential of this specific molecule. This breakthrough provides a foundation for developing therapeutic intervention in diseases driven by aberrant core fucosylation, particularly in cancer.

Allosteric regulation represents a direct and efficient approach to modulating protein function, influencing key biological processes from enzyme catalysis to gene transcription[38]. Our crystal structures revealed an allosteric binding site located between Rossmann domains 1 and 2 of FUT8, characterized by a channel-like hydrophobic pocket capable of accommodating long-chain aliphatic structures. This site exhibits pronounced structural uniqueness when compared to other fucosyltransferases. Our comparative analysis with the crystal structure of FUT9, POFUT1, and AlphaFold2-predicted models of FUT3 highlights that FUT8 possesses a distinct domain architecture[39,40], including an additional Stem region and SH3 domain not found in most other FUT isoforms. Consequently, the spatial arrangement of its Rossmann domains differs substantially (Supplementary Fig. 11), and the allosteric pocket at their interface is structurally unique to FUT8. Critically, although FoldSeek identified POFUT1 and POFUT2 as the top structural neighbors of FUT8's GT-B domain, CAIF exhibited no inhibition against either enzyme. This indicates that despite a similar overall Rossmann domain orientation, the precise topology and chemical environment of the allosteric pocket in FUT8 are not conserved in POFUT1/2. This architectural and micro-environmental distinction likely underlies both FUT8's exclusive function in catalyzing α1,6-core fucosylation and the high selectivity of CAIF, which was designed to complement this unique site. This discovery is particularly significant as it provides a distinct strategy for selectively targeting FUT8 without affecting other members of the fucosyltransferase family.

Covalent inhibition, an emerging strategy in drug discovery, enhances pharmacodynamic efficacy by forming stable covalent bonds with the target, leading to prolonged target engagement[41]. The covalent modification of K216 by CAIF not only enhances the potency and specificity of FUT8 inhibition, but also offers a unique mechanism for sustained enzyme inactivation. The minimal cytotoxicity and significant anti-invasive effects of CAIF align with the biological role of FUT8 in promoting metastasis rather than cell proliferation[6]. The structural insights in this study explain the inhibitory activity of CAIF and provide a blueprint for the designing future allosteric inhibitors targeting FUT8.

We acknowledge that the current CAIF molecule, with its imidazolium salt character and aliphatic chain, primarily serves as a mechanistic probe. Its pharmacokinetic properties and overall drug-likeness necessitate further optimization in future translational studies. When evaluated against the established criteria for high-quality chemical probes[42], CAIF demonstrates key strengths: potent and selective inhibition of FUT8 ($IC_{50} = 5.7\,\mu M$; >114-fold selectivity over FUT3), a well-defined covalent-allosteric mechanism corroborated by structural data, and robust on-target cellular activity. However, consistent with its role as a proof-of-concept tool compound, a matched inactive control molecule has not yet been synthesized. Future work will therefore focus on addressing this, alongside proteomic off-target profiling to fully characterize its specificity, and on medicinal chemistry refinements to improve drug-likeness for in vivo applications.

In summary, this study identifies a previously unrecognized allosteric site on FUT8 and develops CAIF as a potent, selective, and low-toxicity covalent inhibitor. These findings advance our understanding of FUT8 inhibition and establish a framework for the rational design of glycosyltransferase inhibitors. By providing a distinct approach to targeting FUT8, this work not only addresses diseases driven by aberrant glycosylation, particularly cancer, but also sets the stage for future innovations in glycosyltransferase inhibition.

## Methods

### Plasmid construction

The DNA sequences encoding the catalytic domains of human FUT8 (residues 68–575), FUT3 (residues 35–361), and FUT4 (residues 181–530), POFUT1 (residues 24–384), and POFUT2 (residues 22–429) were codon-optimized for expression in human cells and synthesized (Tsingke Biotechnology). Each sequence was cloned into an expression vector featuring an N-terminal 3×Flag tag; FUT8, FUT4, POFUT1, and POFUT2 were cloned into a modified pSectag2A vector, while FUT3 was cloned into the pcDNA3.4 vector. The FUT8 Y220A mutant and the FUT8 (105–575) truncation variant were subsequently generated using the FUT8 (68–575) construct as a template. The DNA sequences encoding human FA9 (residues 93–129) and human thrombospondin-1 TSP1 (residues 379–429) were codon-optimized for expression in *E. coli* and synthesized and cloned into a pET28a vector with an N-terminal His-MBP-TEV tag (Tsingke Biotechnology). The synthesized cDNA sequences and the corresponding deduced mature protein sequences for all constructs are provided in Supplementary Note 2.

## Protein expression and purification

Recombinant proteins were expressed in Expi293F cells (Thermo Fisher Scientific) cultured in suspension in 293F Hi-exp Medium supplemented with L-glutamine (OPM, Cat# AC601501). Cells were maintained at 37 °C, 5% $CO_2$, 50% humidity, and agitated at 110 rpm. Transfection was performed using linear polyethylenimine (PEI-40000, YEASEN) at a concentration of 1 mg/mL. Specific transfection parameters varied by construct: For FUT8 and FUT4, POFUT1, and POFUT2, transfections were performed at a density of $3.0 \times 10^6$ cells/mL using 3 mg of plasmid DNA and 9 mg of PEI per liter of culture. For FUT3, transfections were performed at a density of $2.0 \times 10^6$ cells/mL using 3 mg of plasmid DNA and 4.5 mg of PEI per liter of culture. The FUT8 mutants were expressed using the same parameters as wild-type FUT8. Following transfection, cultures were incubated for 4–5 days before harvesting by centrifugation at $6000 \times g$ for 15 min. The clarified supernatants were loaded onto anti-FLAG M2 affinity resin (GenScript). After extensive washing with a buffer containing 25 mM HEPES-NaOH (pH 7.5) and 150 mM NaCl for FUT3, FUT4, and FUT8, and 25 mM HEPES-NaOH (pH 8.0), 300 mM NaCl and 10 mM $MgCl_2$ for POFUT1 and POFUT2, bound proteins were eluted with the same wash buffer supplemented with 0.2 mg/mL Flag peptide. The eluted proteins were further purified by size-exclusion chromatography (Superdex 200, Cytiva) equilibrated in the purification buffer. Protein concentrations were determined by measuring ultraviolet absorbance at 280 nm a NanoDrop spectrophotometer, with extinction coefficients calculated from the amino acid sequence. Typical yields from the purification process were approximately 1.5 mg/L for FUT8, 0.5 mg/L for FUT3 and the FUT8 mutants, and POFUT2, 0.8 mg/L for FUT4, and 1 mg/L for POFUT1.

Plasmids encoding Human FA9 (residues 93–129) and human TSP1 (residues 379–429) were transform into SHuffle T7 competent *E. coli*. A single colony was inoculated into 100 mL LB medium containing 50 μg/mL kanamycin and incubated at 37 °C, 200 rpm for 6 hours. The culture was then transferred into 1 L fresh LB medium at a 1:200 dilution and incubated at 37 °C until the $OD_{600}$ reached 0.6–0.8. Protein expression was induced by adding 0.5 mM IPTG, followed by incubation at 16 °C, 200 rpm for 20 hours. Cells were harvested by centrifugation at $6000 \times g$ for 15 min at 4 °C, resuspended in lysis buffer (25 mM Tris-HCl, pH 7.5, 150 mM NaCl, 30 mM imidazole) and lysed using a high pressure cell disruptor. The lysate was clarified by centrifugation at $18,000 \times g$ for 30 min at 4 °C. The cleared lysate was loaded onto a pre-equilibrated Ni–NTA affinity chromatography (HisTrap FF column, 5 mL, Cytiva) using an AKTA FPLC system at 4 °C. The resin was washed with 10 column volumes of wash buffer (25 mM Tris-HCl, pH 7.5, 150 mM NaCl, 30 mM imidazole) and then eluted with a linear imidazole gradient (25 mM Tris-HCl, pH 7.5, 150 mM NaCl, 500 mM imidazole). The eluate was further purified by an anion exchange chromatography (HiTrap Q HP column, 5 mL, Cytiva) using an AKTA FPLC system at 4 °C. The resin was washed with 10 column volumes of wash buffer (25 mM Tris-HCl, pH 7.5, 50 mM NaCl) and then eluted with a linear NaCl gradient (25 mM Tris-HCl, pH 7.5, 1 M NaCl). Pure FA9 and TSP1 were aliquoted and stored at −80 °C in 25 mM Tris-HCl, pH 7.5, 150 mM NaCl.

## High-throughput screening (HTS)

High-throughput screening experiments consisted of two steps: the FUT8 reaction and the GDP detection reaction (Supplementary Table 5). In the first step, small molecules (or DMSO) incubated with FUT8 enzyme assay system, which consisted of 10 nM FUT8, 10 μM GDP-Fuc, and 5 μM receptor glycans (a gift from Prof. Ping Wang) in 50 mM HEPES-NaOH (pH 7.5), 100 mM NaCl, 0.01% (v/v) Triton X-100, and 0.2% (w/v) BSA. Two compound libraries from Selleck Chemicals were screened: the U.S. Food and Drug Administration (FDA)-approved Drug Library (2024 compounds) and the Express-Pick Library (4224 compounds). The former was tested at a final

concentration of 100 μM, and the latter at 10 μM. Due to the lack of a highly potent FUT8 inhibitor, the experimental system without FUT8 was used as a surrogate positive control, while the DMSO group served as the negative control. The reaction system was prepared in a 96-well PCR plate (19.8 μL/well), followed by the addition of 0.2 μL of either the test compounds or DMSO using a Gryphon automated liquid handler (ART Technologies), resulting in a final DMSO concentration of 1% (v/v). The plate was sealed and incubated at 37 °C for 1 hour. After incubation, 10 μL of the reaction mixture from each well was transferred to a white 384-well plate (Corning) and mixed with an equal volume (10 μL) of GDP detection reagent. The plate was then protected from light and incubated at room temperature for 1 hour. GDP production was quantified by luminescence measurement using an EnVision® Multimode Plate Reader (PerkinElmer).

The quality of the HTS assay was assessed by calculating the Z′-factor, a measure of assay robustness in the absence of any test compounds, as described by Zhang et al.[43]. The formula used was: $Z' = 1 − [3 \times (\sigma_p + \sigma_n) / |\mu_p − \mu_n|]$, where $\mu_p$ and $\mu_n$ are the means, and $\sigma_p$ and $\sigma_n$ are the population standard deviations (calculated using the STDEV.P function in Excel) of the positive (p) and negative (n) controls, respectively. A Z′-factor value of 0.5 is considered indicative of an excellent assay[43]. All screening data were normalized against the plate control values. Compounds that reduced FUT8 enzymatic activity by more than 40% (i.e., normalized signal <0.6) were selected for further validation.

## TRANSCREENER GDP FI assay

The experiment was conducted in a two-step procedure. In the initial step, both the experimental system setup and control group configuration were maintained identical to those established in the HTS protocol: 10 nM FUT8, 10 μM GDP-Fuc, and 5 μM receptor glycans in 50 mM HEPES-NaOH (pH 7.5), 100 mM NaCl, 0.01% (v/v) Triton X-100, and 0.2% (w/v) BSA. The reactions were carried out in eight-tube strips, each containing a total reaction volume of 25 μL. GDP production was quantified using the TRANSCREENER® GDP FI Assay (BellBrook Labs) according to the manufacturer's protocol. Briefly, 20 μL aliquots of the reaction mixture were transferred to a black 96-well plate (Corning) and combined with 20 μL of detection reagent mixture containing GDP antibody-IrDye® Qc-1 and GDP Alexa594 Tracer in Stop & Detect Buffer B (1:1 ratio). After thorough mixing, the plate was incubated at room temperature for 60 minutes. Fluorescence intensity was measured at $\lambda_{ex}$ 580 nm and $\lambda_{em}$ 620 nm. All experiments were performed in quadruplicate.

## Validation of CAIF selectivity in the GDP-detection system

To verify that CAIF does not interfere with the GDP-detection system, we performed control experiments in a total reaction volume of 20 μL (10 μL assay mixture + 10 μL detection reagent). The assay mixture contained 10 μM GDP and CAIF at concentrations ranging from 0 to 1 mM in assay buffer (50 mM HEPES-NaOH, pH 7.5, 100 mM NaCl, 0.01% (v/v) Triton X-100, 0.2% (w/v) BSA). After incubation, 10 μL of GDP-detection reagent was added (1:1 ratio) in white 384-well plates (Corning), followed by 1-hour incubation in the dark at room temperature. Luminescence signals were measured using either an EnVision® Multimode Plate Reader (PerkinElmer) or a GloMax Multi-detection system (TECAN). Data analysis was performed using linear regression in GraphPad Prism 7, with quadruplicate measurements for each condition. These experiments confirmed that CAIF at all tested concentrations showed no detectable effect on the GDP-detection system (Supplementary Fig. 9a), demonstrating its specific inhibition of FUT8.

## Dose-response enzyme activity inhibition assays

The enzymatic activities of FUT8, FUT3, FUT4, POFUT1, and POFUT2 were assessed using the GDP-Glo™ Glycosyltransferase Assay (Promega), which quantifies the GDP released during the

glycosyltransferase reaction. Compounds were solubilized in DMSO to generate 200× stock solutions, which were serially diluted to generate eleven-point concentration curves. Each dilution was added to the assay mixture at a 1:100 ratio (v/v). The enzyme-compound mixtures were pre-incubated on ice for 2 hours. This extended pre-incubation protocol was employed in accordance with established practices for characterizing covalent inhibitors, particularly when the determination of $k_{inact}/K_I$ is experimentally challenging[44,45]. The reaction was initiated by combining the pre-incubated mixture with an equal volume of substrate solution, yielding final reaction conditions of 10 nM FUT8 or 25 nM FUT3 or 50 nM FUT4 or 20 nM POFUT1 or 20 nM POFUT2, 0.5% (v/v) DMSO, and the following components in their respective buffers: FUT8: 10 µM GDP-Fuc, 5 µM acceptor glycan in 50 mM HEPES-NaOH (pH 7.5), 100 mM NaCl, 0.01% (v/v) Triton X-100, and 0.2% (w/v) BSA. FUT3: 10 µM GDP-Fuc, 50 µM LacNAc in 50 mM HEPES-NaOH (pH 7.5), 100 mM NaCl, 0.01% (v/v) Triton X-100, 0.2% (w/v) BSA and 1 mM $MnCl_2$. FUT4: 10 µM GDP-Fuc, 50 µM N-acetyllactosamine (Tokyo Chemical Industry) in 50 mM HEPES-NaOH (pH 7.5), 100 mM NaCl, 0.01% (v/v) Triton X-100, and 0.2% (w/v) BSA. POFUT1: 10 µM GDP-Fuc, 10 µM FA9 in 50 mM HEPES-NaOH (pH 7.5), 100 mM NaCl, 0.01% (v/v) Triton X-100, and 0.2% (w/v) BSA, 1 mM $MgCl_2$. POFUT2: 10 µM GDP-Fuc, 10 µM TSP1 in 50 mM HEPES-NaOH (pH 7.5), 100 mM NaCl, 0.01% (v/v) Triton X-100, and 0.2% (w/v) BSA, 1 mM $MgCl_2$. Reactions were incubated at 37 °C for 1 hour (FUT8, FUT4, POFUT1, POFUT2) or 2 hours (FUT3) before being stopped by adding an equal volume of GDP-Glo detection reagent. After a further 1-hour incubation at room temperature, luminescence was measured using an EnVision® Multimode Plate Reader (PerkinElmer) or a GloMax® Discover Microplate Reader (Promega). Dose-response curves and half-maximal inhibitory concentration ($IC_{50}$) values were calculated using nonlinear regression analysis in GraphPad Prism 7.

## The cell lines

The human malignant melanoma cell lines A375 and GAK, human cervical adenocarcinoma cell line HeLa, and human embryonic kidney cell line HEK293T cell lines were purchased from the American Type Culture Collection and cultured in Dulbecco's Modified Eagle Medium (DMEM; Invitrogen) supplemented with 10% fetal bovine serum (FBS; Gibco) in 6 cm culture dishes. Their selection was based on their expression of FUT8 and their relevance to studying cancer biology and core fucosylation[6,46]. For specific experiments, cells were seeded at appropriate densities in various culture vessels, including 6 cm dishes, 12-well plates, or 96-well plates, depending on experimental requirements.

## Cell viability assessment

Cell viability was evaluated using the Cell Counting Kit-8 (CCK-8) assay according to the manufacturer's instructions. Briefly, cells were seeded into transparent 96-well plates at a density of $5 \times 10^3$ cells per well in 100 µL of complete medium and cultured overnight under standard culture conditions (37 °C, 5% $CO_2$). The next day, cells were treated with serially diluted compounds, while the negative control group received an equivalent volume of DMSO. Four replicate wells were used for each treatment condition. Blank wells containing culture medium with the corresponding concentrations of compounds (without cells) were included to correct for non-specific background absorbance. After 72 hours of compound exposure, 10 µL of CCK-8 reagent (APExBIO, Cat# K1018) was added to each well. Plates were subsequently incubated for an additional 4 hours at 37 °C. Absorbance was measured at 450 nm using an EnVision® Multimode Plate Reader (PerkinElmer), and data were analyzed using GraphPad Prism 7 software. All experiments were performed in quadruplicate.

## Lectin fluorescence-based cell imaging

GAK or A375 cells were seeded into black, clear-bottom 96-well plates (Greiner) at a density of $5 \times 10^3$ cells per well and cultured overnight

under standard conditions (37 °C, 5% $CO_2$). After attachment, cells were treated with CAIF at concentrations ranging from 0 to 300 µM (from a 100× DMSO stock diluted in medium) for 72 hours. An equivalent volume of DMSO was added to the control wells. Following treatment, cells were washed three times with PBS and fixed with 4% paraformaldehyde (PFA) at room temperature for 20 min. After fixation, cells were blocked with 5% (w/v) BSA in PBS at room temperature for 2 h. For fucosylation detection, FITC-conjugated AAL or LCA lectins were diluted in PBS containing 2.5% (w/v) BSA to final concentrations of 2 µg/mL and 5 µg/mL, respectively, and incubated with the cells for 1 h at room temperature in the dark. After lectin staining, cells were washed and counterstained with Hoechst 33342 (1 µg/mL in PBS) for nuclear visualization. Fluorescence imaging was performed using the Operetta High-Content Imaging System (PerkinElmer). Image analysis was conducted using the Multiwavelength Cell Scoring module in MetaXpress software (Molecular Devices). The average integrated fluorescence intensity was quantified and normalized to cell number per field. Four fields per well were analyzed, with a minimum of four replicate wells per condition. Fucosylation levels were normalized to DMSO-treated controls and analyzed using GraphPad Prism 7 software.

## Flow cytometry

A375 cells were seeded into 6-well plates at a density of $1 \times 10^5$ cells per well and maintained in DMEM supplemented with 10% FBS and 1% Penicillin-Streptomycin. Cells were treated for 72 hours with either DMSO control, CHX, or NH125. Following treatment, cells were harvested and stained with fluorescein-conjugated LCA. Fluorescence intensity was quantified using a Guava EasyCyte™ 8HT flow cytometer, and data were analyzed using FlowJo software.

## In vitro invasion assay

Cell invasion was assessed using Corning® Transwell® 24-well plates (Sigma-Aldrich, Cat# CLS3421). A375 or HeLa cells were pretreated with varying concentrations of CHX, NH125, or CAIF for 48 hours prior to the assay. For migration analysis, $5 \times 10^4$ cells in serum-free medium were seeded into the upper chamber pre-coated with Matrigel matrix (Corning, Cat# 356234), while the lower chamber contained complete medium with 20% FBS as a chemoattractant. Following 24-hour incubation at 37 °C, migrated and invaded cells were stained with 5 µg/mL Calcein AM (Sangon Biotech, China) for 30 minutes at 37 °C. Cell invading was quantified by counting five random fields per insert using fluorescence microscopy.

## Structural determination of FUT8-inhibitor complexes

To elucidate the molecular mechanisms of FUT8 inhibition, we expressed and purified the human FUT8 (105-575) for co-crystallization studies with candidate inhibitors (NH125, CHX, SSO, and CAIF). High-quality co-crystals were obtained for all FUT8-inhibitor complexes. X-ray diffraction data were collected at the BL19U1 beamline at the National Facility for Protein Science in Shanghai, and beamlines BL02U1 and BL10U1 of the Shanghai Synchrotron Radiation Facility[47]. The data were processed using beamline-integrated software packages (autoPROC, Propoise, and xia2) for unit cell determination, space group assignment, data integration, and scaling[48,49]. Molecular replacement was performed using the Phaser program in the CCP4 suite with the apo-FUT8 structure (PDB: 6X5S) as the search model[22,50]. Iterative model building and refinement were conducted using Coot and Phenix[51,52]. The data collection and model statistics are given in Supplementary Tables 3 and 4.

## Molecular dynamics (MD) simulation and analysis

MD simulations were performed using GROMACS 2024.2 to study the dynamics of both apo-FUT8 and the FUT8–NH125 complex[53]. The CHARMM36-July2022 force field was employed for its accurate

parametrization of proteins and small molecule ligands[54], in conjunction with the TIP3P water model[55]. Each system contained a FUT8 dimer solvated in a dodecahedral periodic box with a minimum distance of 1.0 nm between the protein and the box boundary. The system was neutralized with $Na^+$ ions (Supplementary Table 6). The protonation states of all ionizable residues were set to their default values at pH 7.0. Energy minimization was carried out using the steepest descent algorithm until the maximum force was below 10 kJ/mol. The minimized system was then used as the starting point for all subsequent independent replicates. The equilibration protocol consisted of two phases: For each independent replicate, a 100 ps NVT simulation was first performed at 300 K using the V-rescale thermostat (tau_t = 0.1 ps) with initial velocities generated from a Maxwell-Boltzmann distribution at 300 K using a random seed (gen_seed = −1). This was followed by a 100 ps NPT simulation at 300 K and 1.0 bar using the C-rescale barostat (tau_p = 2.0 ps). During equilibration, the protein and ligand heavy atoms were restrained to their initial positions using a harmonic force constant of 1000 kJ/mol/nm².

For all simulations, the Verlet cutoff scheme was employed. Short-range nonbonded interactions were treated with a cutoff of 1.2 nm. Long-range electrostatics were handled using the Particle Mesh Ewald (PME) method with a Fourier spacing of 0.16 nm and an interpolation order of 4. The van der Waals interactions were modified with a force-switch function, smoothly shifting the potential to zero between 1.0 and 1.2 nm. The neighbor list was updated every 20 steps. Bonds involving hydrogen atoms were constrained using the LINCS algorithm (lincs_order = 4, lincs_iter = 1), allowing for a 2-fs integration time step. Production simulations were then conducted for 100 ns under NPT conditions using the Parrinello–Rahman barostat. All restraints were removed for the production phase. Thus, for each system (apo and NH125-bound), three truly independent replicates were generated, differing only in their initial random velocity assignments from the start of the NVT equilibration phase. All analyses were performed on the production-phase trajectories from each independent replicate after confirming stability via root-mean-square deviation (RMSD) analysis. For each replicate, a stable trajectory segment was selected for subsequent calculations to characterize the dynamic behavior of FUT8 in both the inhibitor-bound and apo states.

## Reporting summary
Further information on research design is available in the Nature Portfolio Reporting Summary linked to this article.

## Data availability
Unless otherwise stated, all data supporting the results of this study can be found in the article, supplementary, and source data files. Previously published structural data used in this study are available under accession codes: 6X5S, 6TKV, 8D0O, and 5UX6. Structural data generated in this study, including atomic coordinates and structure factors, have been deposited in the Protein Data Bank under accession numbers 9L62, 9L63, 9L64 and 9L65. The initial and final coordinate files for the molecular dynamics simulations are provided as Supplementary Data. Source data are provided with this paper.

## Code availability
For the principal component analysis in Fig. 3d, the associated Python script is publicly available at Zenodo [https://doi.org/10.5281/zenodo.18067084]. All other results were obtained using standard analysis tools without custom code.

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

## Acknowledgements

This work is supported by the National Natural Science Foundation of China grants 92578104, 22322706, 82441039, and 22521104, Strategic Priority Research Program of the Chinese Academy of Sciences grant No. XDB1060000, the National Key R&D Program of China grant No. 2022YFA1304700, and the Central Guiding Local Science and Technology Development Fund Projects No. 2025ZY01043. We acknowledge Prof. Wenjun Tang for his support in organic synthesis, Prof. Peng Xu for his valuable discussions, and Prof. Ping Wang for kindly supplying receptor glycans. We thank the Chemical Biology Core Facility in CEMCS, CAS, Lin Qiu and Jiayu Wu from the Institutional Center for Shared Technologies and Facilities of SINH, CAS, for technical assistance. We also acknowledge the staff at the BL19U1 beamline (National Facility for Protein Science, Shanghai) and the BL10U2/BL02U1 beamlines (Shanghai Synchrotron Radiation Facility) for their support during X-ray diffraction data collection and analysis.

## Author contributions

B.Y., J.W., and P.F. conceived and supervised the project; J.J., D.H., M.K., J.Q., and G.Y. performed the investigation and collected data; J.J., D.H., M.K., and P.F. conducted the formal analysis and wrote the original draft; B.Y., J.W., and P.F. reviewed and edited the manuscript. All authors discussed the results and commented on the manuscript. All authors read and approved the final manuscript.

## Competing interests

The authors declare no competing interests.
