## [Transparent Peer Review file · Nature Communications]

Exploiting human fucosyltransferase 8 allostery with a covalent inhibitor for core fucosylation suppression

Corresponding Author: Professor Pengfei Fang

Version 0:

Reviewer comments:

Reviewer #1

(Remarks to the Author)

Overall assessment

This is a well-written manuscript describing a wide-ranging study, encompassing protein expression and purification, enzyme assays including high-throughput screening, X-ray crystallography, computational studies and chemical synthesis. The study described is a significant advance on those in the literature, showing that targeted inhibition of the enzyme can be achieved with selectivity against other fucosyl transferases. The authors have developed a covalent allosteric inhibitor. I am not convinced that such compounds will be druggable but nevertheless this manuscript is an important proof-of-concept study.

The kinetic characterization of the synthesized compound is insufficient since potency has been measured using an IC50 value. It has long been recognized that this is inappropriate for covalent inhibitors since the derived value is dependent on the exposure time of the target to the inhibitor. Although there is a suggestion in the literature that IC50 values can be used with long incubation times are used (Thorarensen, et al., *Bioorg. Med. Chem.* 29 (2021) 115865), the accepted way to do this is to measure the rate of inactivation at several inhibitor concentrations and derive the kinactivation/KI value for their compounds. The authors are referred to the textbook by Robert Copeland, *Evaluation of enzyme inhibitors in drug discovery: A guide for medicinal chemists and pharmacologists*, 2nd ed. Wiley.

As the authors are using a coupled assay, they need to provide evidence that inhibition of the target is happening and that CAIF is not inhibiting the coupling enzymes (control experiments).

There are some additional details that need to be added to the methods and elsewhere as detailed here:

- Buffers need to be properly defined including the counter-ion. HEPES is presumably HEPES-NaOH. % concentrations need to be specified (presumably these are % v/v for Triton X-100 and % w/v for BSA).
- For the HTS and other enzyme assay, what was the final DMSO concentration? Was this consistent throughout? How long were compounds pre-incubated with enzyme before measuring activity? This is a key consideration for covalent inhibitors as inhibition will be dependent on pre-incubation time.
- Please confirm that Z' was calculated using the population standard deviations (the paper by Zhang et al., 1999 used population standard deviations although this has been subsequently criticised).
- Figure 1D. The dose-response curve must use log10 molar drug concentrations. The values should be quoted using pIC50 ± SEM, since the distribution of IC50 estimates is skewed and therefore non-parametric. The dose-response curve for mitoxantrone needs to be repeated with higher drug concentrations because there is insufficient plateau on the right-hand side of the curve. The legend says "Experiments were performed in quadruplicate, with error bars representing the standard deviation (n = 4, mean ± SD)." Presumably the displayed dose-response curve is a representative independent repeat, with data collected at each drug concentration more than once. N numbers refer to independent repeats. Identical comments apply to the dose-response curves for figures 4 and 5 and those in the supplementary information.

- Figure S2d. What statistical correction was used for this data (Bonferroni?). Details of the statistical methods need to be given in the supplementary information.

- Both synthesized compounds (compound 2 and CAIF) are novel. Novel compounds generally require 5 pieces of characterization data; why is there no ¹³C data and spectra for CAIF? The authors should report ¹H and ¹³C spectra, HRMS and IR as a minimum.

- There is one error in the supplementary crystallographic Table 2 - Structure FUT8-CAIF (9L65), the authors have entered '7824' solvent molecules were identified in the structure - which need to be corrected.

Reviewer #2

(Remarks to the Author)

Please see attached document

Reviewer #3

(Remarks to the Author)

The manuscript presents a novel allosteric site on FUT8 and the development of CAIF, a covalent inhibitor, through structure-based drug design. The work integrates high-throughput screening, crystallography, and cellular assays to demonstrate CAIF's efficacy in suppressing core fucosylation and cancer cell invasion. While the structural and mechanistic insights are commendable, critical gaps in validation, selectivity, and translational relevance significantly weaken the study's impact. Below are detailed concerns from this reviewer

Major Concerns

1. CAIF's low cytotoxicity and anti-invasive effects are demonstrated only in cellular assays. Without animal model data (e.g., tumor growth/metastasis suppression, pharmacokinetics, or toxicity profiles), the therapeutic potential of CAIF remains speculative. The absence of *in vivo* experiments undermines the clinical relevance of the findings.
2. FUT8 belongs to a family of fucosyltransferases with shared structural motifs. While the authors claim CAIF's selectivity via its allosteric site, no data are provided to confirm that CAIF does not inhibit other FUT enzymes. Selectivity assays against FUT1–FUT7/9 or homology modeling to compare binding sites are essential to validate specificity claims.
3. The unresolved loops (e.g., residues 77–104) and conformational changes in the 215NKG217215NKG217 loop are not rigorously analyzed. A correlation between B-factors (crystallography) and RMSF values (MD simulations) would clarify dynamic contributions to allostery.
4. The MD-derived free energy plots (Fig. 3c–f) lack quantitative descriptors (e.g., energy barriers, metastable states). Markov state modeling (MSM) or residue-level hydrogen bonding/fluctuation analyses would strengthen mechanistic insights into how NH125/CAIF binding alters FUT8 dynamics.
5. The K216-CAIF covalent interaction lacks orthogonal validation (e.g., mass spectrometry, activity assays under reducing conditions). Mutagenesis (K216A) is also required to confirm its functional role in inhibitor binding.
6. As a covalent inhibitor, CAIF's potential for off-target modifications is not addressed. Proteomic profiling or kinetic assays with unrelated lysine-dependent enzymes are needed to assess specificity.
7. CAIF's long aliphatic chain raises concerns about solubility and bioavailability. Pharmacokinetic studies (absorption, distribution, metabolism, excretion) are absent and critical for translational evaluation.
8. The proposed steric clash between Y220 and Y250 (Fig. 3a) lacks experimental support. Mutating Y220 to alanine and testing inhibitor efficacy would clarify its role in GDP ribose binding and allosteric inhibition.
9. The dimeric state of FUT8 under physiological conditions and its impact on inhibitor binding (e.g., cooperative effects) are not discussed. Disruption of the dimer interface (via mutagenesis) could reveal its functional relevance.

Minor

1. Fig. 1a: Labels for positive/negative controls and hits are unclear; a legend is needed.
2. Fig. 3c–f: Axes for free energy landscapes are unlabeled; quantitative energy values should be included.
3. Tables: IC₅₀ values for inhibitors are scattered; a consolidated table in supplementary materials would improve clarity.

Reviewer #4

(Remarks to the Author)

Reviewer #5

(Remarks to the Author)

The manuscript reports the discovery of a novel allosteric site on FUT8 and the development of a low-toxicity covalent inhibitor, CAIF (stearic acid-N-hydroxysuccinimide ester-dimethylimidazolium bromide), through structure-based drug design. High-throughput screening and crystallographic studies revealed that small molecules, such as NH125, bind to a

channel-like allosteric pocket, inducing conformational changes that disrupt FUT8 activity. Leveraging these insights, they designed CAIF to covalently target lysine K216 within the allosteric site of FUT8. CAIF exhibits minimal cytotoxicity and significantly inhibits core fucosylation and cancer cell invasion in cellular assays. This work establishes CAIF as a lead compound for further optimization and development, offering a framework for targeting glycosyltransferases through allosteric and covalent inhibition strategies.

The manuscript is well written and has significant results and therefore can be considered for publication in Nature Communication after the responses to following comments are incorporated.

1. The authors claimed CAIF as a specific FUT8 inhibitor based on its interaction with allosteric site. However, as all 12 isoforms of FUTs share significant structural similarity and thus may possess similar allosteric sites as well. In this context, how authors hypothesized that CAIF will specifically inhibit FUT8, and will not affect the other isoforms of FUTs.
2. Authors mentioned that CAIF exhibits minimal cytotoxicity, while significantly inhibiting the core fucosylation, and cancer cell invasion in cellular assays. This statement appears to be self-contradictory, and thus needs further explanation.
3. Various pro- and eukaryotic cell lines are mentioned in the manuscript. Authors needs to explain the selection criteria of these cell lines with respect to FUT8 in the manuscript. The details of these cell lines also needs to be mentioned.
4. Since, FUT8 is a bi-substrate enzyme and has two distinct donor and acceptor substrate binding sites. The 2nd and 3rd paragraph in the introduction section should be merged and reduced in a single paragraph. While a new paragraph needs to be added highlighting the key structural details of both the binding sites of FUT8.
5. Authors have not provided any details regarding the protein (FUT8) expression and purification in the Results Section. This needs to be incorporated in the revised manuscript.
6. They also need to mention the level of purity and yield of FUT8 protein.
7. In in-vitro FUT8 inhibition assays, IC50 values are mentioned without Standard Error of Mean (S.E.M). Authors need to incorporate the S.E.M with all mentioned IC50 values.
8. 1H and 13C -NMR data of compound 2 is provided in supplementary materials. However, it is not clear which compound is compound 2?
9. 13C-NMR chemical shift data of the CAIF is missing.
10. Citations of figures need to be checked by the authors.

Reviewer #6

(Remarks to the Author)

Version 1:

Reviewer comments:

Reviewer #1

(Remarks to the Author)

I am satisfied with the revisions made by the authors. However, there are four points of clarification needed in the following paragraph: "The screening quality was assessed by calculating the Z'-factor using the formula: $Z' = 1 - [3 \times (s_p + s_n) / |\mu_p - \mu_n|]$, where μ_p and μ_n are the means, and s_p and s_n are the sample standard deviations (with Bessel's correction, $n-1$ in the denominator) of the positive (p) and negative (n) controls, respectively. Calculations were performed using the STDEV.S function in Excel. A Z'-factor value > 0.5 was considered indicative of a robust screening system. Experimental data were normalized against control values, and compounds demonstrating normalized values < 0.6 were selected for further validation."

1. Zhang et al., 1999 J. Biomol. Screen. 4, 67-73 used the population standard deviations in the calculation of Z' (this has been criticized on statistical grounds, but nevertheless Z' should be calculated as described in this paper).
2. The requirement for Z' to be equal to or greater than 0.5 is given in Table 1 of Zhang et al. 1999 (above). The authors may wish to cite this paper in the sentence "A Z'-factor value > 0.5 was considered indicative of a robust screening system."
3. Technically, Z' is a measure of assay performance in the absence of any library compounds (the corresponding parameter in the presence of library compounds is Z).
4. The final sentence in this paragraph is ambiguous in that it appears to relate to Z'. I think the authors are saying that compounds which reduced enzymatic activity by 40% or more were selected for follow up. This needs to be updated to avoid ambiguity.

Reviewer #2

(Remarks to the Author)

Thank you for the revised manuscript and for thoroughly addressing all referees' comments. I think this version is significantly strengthened and easier to read. Below are suggestions to further improve the manuscript.

Regarding my main concern on specificity claims, authors have shown experimentally that their inhibitor did not inhibit FUT3 and FUT4 although those are functional and not structural homologs of FUT8. Authors may consider screening the human proteome computationally (such as using FoldSeek) for proteins that are structural homologs of FUT8 and test compound

specificity on those.

The addition of the FEL adds an interesting aspect to the potential mechanism of action of the designed inhibitors and could be useful in further efforts to optimize the binder into a drug. The authors may consider adding a structural representative of each metastable conformational state to shed some light on the residue level interactions that were altered when the protein is bound to NH125.

Reviewer #3

(Remarks to the Author)

The authors have comprehensively addressed the majority of the concerns raised during the initial review. The revisions significantly strengthen the manuscript, particularly through the addition of critical experimental data and enhanced mechanistic insights.

Reviewer #5

(Remarks to the Author)

This study presents a highly significant advancement in the field of glycosylation-targeted therapeutics by identifying an allosteric pocket in FUT8 and designing the first low-toxicity covalent inhibitor (CAIF) that selectively disrupts FUT8 activity. The work is important because FUT8-mediated core fucosylation is a critical driver of cancer progression, immune evasion, and drug resistance, yet the lack of selective inhibitors has been a major obstacle due to the shared substrate specificity of fucosyltransferases. By combining structural insights, high-throughput screening, and crystallographic validation, the authors not only provide evidence of allosteric modulation of FUT8 but also establish CAIF as a lead compound with promising biological efficacy in inhibiting cancer cell invasion. Beyond FUT8, this approach offers a broadly applicable framework for designing allosteric and covalent inhibitors against glycosyltransferases, thereby opening new therapeutic avenues in oncology and immune modulation.

The manuscript presents significant results and has justified and supported their data with various techniques. All the suggested revisions have been incorporated. The support from additional experimental data significantly improves the manuscript. Hence, the manuscript is acceptable to be published in the journal in its current form.

Reviewer #6

(Remarks to the Author)

Version 2:

Reviewer comments:

Reviewer #1

(Remarks to the Author)

The authors have incorporated all of the required changes from the latest round of review. I have no further suggested changes.

Reviewer #2

(Remarks to the Author)

Thank you for the revised manuscript and for following through with all the suggestions. I'm glad the additional experiments added new, interesting insights to the story. I think the manuscript has significantly been strengthened and publishable in its current form.

Response to Reviewers' Comments

Manuscript ID: NCOMMS-25-06314-T

Manuscript Title: Exploiting human fucosyltransferase 8 allostery with a covalent inhibitor for core fucosylation suppression

Reviewer #1:

Overall assessment

This is a well-written manuscript describing a wide-ranging study, encompassing protein expression and purification, enzyme assays including high-throughput screening, X-ray crystallography, computational studies and chemical synthesis. The study described is a significant advance on those in the literature, showing that targeted inhibition of the enzyme can be achieved with selectivity against other fucosyltransferases. The authors have developed a covalent allosteric inhibitor. I am not convinced that such compounds will be druggable but nevertheless this manuscript is an important proof-of-concept study.

Response: We sincerely thank the reviewer for the thoughtful evaluation of our manuscript and the recognition of its significance as "an important proof-of-concept study".

The kinetic characterization of the synthesized compound is insufficient since potency has been measured using an IC₅₀ value. It has long been recognized that this is inappropriate for covalent inhibitors since the derived value is dependent on the exposure time of the target to the inhibitor. Although there is a suggestion in the literature that IC₅₀ values can be used with long incubation times are used (Thorarensen, et al., *Bioorg. Med. Chem.* 29 (2021) 115865), the accepted way to do this is to measure the rate of inactivation at several inhibitor concentrations and derive the $k_{\text{inact}}/K_{\text{I}}$ value for their compounds. The authors are referred to the textbook by Robert Copeland, *Evaluation of enzyme inhibitors in drug discovery: A guide for medicinal chemists and pharmacologists*, 2nd ed. Wiley.

Response: We fully agree with the reviewer that the derivation of $k_{\text{inact}}/K_{\text{I}}$ is the gold standard for covalent inhibitors, as it provides a time-independent measure of potency. However, our compound's allosteric covalent mechanism introduces extra experimental challenges: unlike

active-site covalent inhibitors, allosteric inhibitors cannot be competitively displaced by high concentrations of substrate, making it hard to quench the covalent reaction using regular methods. Therefore, we attempted to quench the reaction with excess lysine. However, due to the complex nature of the FUT8 assay system and unknown quenching efficiency, and despite extensive optimization attempts, we were unable to establish a stable k_{inact}/K_I assay. As a reference, we have included one representative dataset below showing time-dependent inactivation at four inhibitor concentrations. From this dataset, we estimated approximate kinetic parameters: $k_{\text{inact}} = 3.7 \text{ min}^{-1}$, $K_I = 24.5 \text{ }\mu\text{M}$, yielding $k_{\text{inact}}/K_I = 0.15 \text{ }\mu\text{M}^{-1}\cdot\text{min}^{-1}$. The data indicate that covalent modification reaches near-plateau within 2 minutes.

Figure for review. Determination of Covalent Inhibition Kinetics for CAIF. (A)

Time-dependent inactivation of FUT8 by CAIF at varying concentrations. Residual enzyme activity (%) is plotted against incubation time (minutes) for four inhibitor concentrations (1, 10, 30, and 100 μM). Error bars represent SD of four technical repeats.

(B) Dependence of observed inactivation rates (k_{obs}) on CAIF concentration.

In our study, we determined the IC_{50} values for CAIF inhibitors using a 2-hour pre-incubation period to ensure complete covalent modification before measuring residual enzyme activity. This approach aligns with the reviewer's recommended literature, where prolonged incubation IC_{50} values are accepted for covalent inhibitors when kinetic assays are impractical.

We have now explicitly detailed this IC_{50} determination protocol in the Methods section of our revised manuscript (*Page 18, Lines 18–21*), with appropriate references to the reviewer's recommended literature.

As the authors are using a coupled assay, they need to provide evidence that inhibition of the target is happening and that CAIF is not inhibiting the coupling enzymes (control experiments).

Response: We appreciate the reviewer's insightful suggestion regarding the need to demonstrate that CAIF specifically inhibits the target enzyme FUT8 rather than the coupling enzymes in our assay system.

To address this important concern, we have now performed additional control experiments where we directly incubated varying concentrations of CAIF with the detection system (containing the coupling enzymes and GDP substrate, but without FUT8). The results (*newly added **Supplementary Figure 8a***) clearly demonstrate that CAIF shows no inhibitory effect on the coupling enzymes up to 1 mM, confirming that the observed inhibition in our main assays is indeed due to specific targeting of FUT8. This control experiment strengthens our conclusion about CAIF's selectivity and addresses the potential confounding factor raised by the reviewer. We thank the reviewer for this valuable suggestion, which has helped us improve the rigor of our mechanistic studies.

Supplementary Figure 8 | Selectivity of CAIF and its dependence on the allosteric site of FUT8. a) CAIF titration against a negative control assay, showing no inhibitory activity.

There are some additional details that need to be added to the methods and elsewhere as detailed here:

- Buffers need to be properly defined including the counter-ion. HEPES is presumably HEPES-NaOH. % concentrations need to be specified (presumably these are % v/v for Triton X-100 and % w/v for BSA).

Response: As suggested, these clarifications have been incorporated into the Methods section (*Page 16, Lines 7, 20; Page 17, Line 18; Page 18, Lines 3, 25, 26, 29*) to ensure full reproducibility.

- For the HTS and other enzyme assay, what was the final DMSO concentration? Was this consistent throughout? How long were compounds pre-incubated with enzyme before measuring activity? This is a key consideration for covalent inhibitors as inhibition will be dependent on pre-incubation time.

Response: We have now clarified the following points in the revised manuscript: In the high-throughput screening (HTS), the final DMSO concentration was 1% (v/v) (*Page 16, Line 29*). For the concentration-dependent inhibition assays, a lower final DMSO concentration of 0.5% (v/v) was used to minimize any potential nonspecific solvent effects on enzyme activity at higher compound concentrations (*Page 18, Line 23*). All control groups were matched to the same DMSO concentration as the corresponding test compounds within each experiment.

Compounds were pre-incubated with FUT8 on ice for 2 hours before activity measurement which is a sufficient time for covalent modification. These details have been added to the Methods section (*Page 18, Line 19*). We appreciate the reviewer's emphasis on these parameters, which are indeed pivotal for interpreting covalent inhibition data.

- Please confirm that Z' was calculated using the population standard deviations (the paper by Zhang et al., 1999 used population standard deviations although this has been subsequently criticised).

Response: We appreciate the reviewer's careful attention to the statistical notation in our manuscript. We sincerely regret the incorrect use of the population standard deviation symbol (σ) in our original Methods section, which may have caused confusion. We confirm that all calculations were actually performed using sample standard deviations (s) with Bessel's correction (denominator = n-1), which is the appropriate approach for our experimental data.

We have now corrected the Methods section (*Page 17, Lines 6–9*) to explicitly state this

correction, now clearly specifying that standard deviations were calculated as sample statistics using Excel's STDEV.S function.

- Figure 1D. The dose-response curve must use log₁₀ molar drug concentrations. The values should be quoted using pIC₅₀ ± SEM, since the distribution of IC₅₀ estimates is skewed and therefore non-parametric. The dose-response curve for mitoxantrone needs to be repeated with higher drug concentrations because there is insufficient plateau on the right-hand side of the curve. The legend says “Experiments were performed in quadruplicate, with error bars representing the standard deviation (n = 4, mean ± SD).” Presumably the displayed dose-response curve is a representative independent repeat, with data collected at each drug concentration more than once. N numbers refer to independent repeats. Identical comments apply to the dose-response curves for figures 4 and 5 and those in the supplementary information.

Response: We thank the reviewer for these critical and constructive suggestions regarding the analysis and presentation of our dose-response data. We have revised the figures and legends accordingly. In accordance with the recommendation, the x-axes of all dose-response curves (Figures 1d, 4b, 5b, and relevant Supplementary Figures) now represent log₁₀ molar drug concentration. The inhibitory potency is now quantified and reported as pIC₅₀ ± SE to account for the skewed distribution of IC₅₀ estimates, as suggested.

For mitoxantrone, we repeated the experiment with the maximum tested concentration increased to 3 mM. While the new data (*newly added Supplementary Figure 2c*) show a clearer trend towards a plateau, full saturation was not achieved even at this highest soluble concentration. Therefore, we conservatively labeled the activity as pIC₅₀ < 4.16 in original **Fig. 1d** and pIC₅₀ < 3.82 in new **Supplementary Figure 2c** and revised the text to state “while mitoxantrone was less potent” without claiming a definitive value (**Page 6, Line 11**).

We have meticulously updated all relevant figure legends to unambiguously define the nature of the data and error bars. For example, the legend for **Fig. 1d** now reads: “Data points show the mean ± SD of quadruplicate technical repeats from one representative independent experiment. The dose-response curve from another independent replicate is provided in

Supplementary Figure 2.”

- Figure S2d. What statistical correction was used for this data (Bonferroni?). Details of the statistical methods need to be given in the supplementary information.

Response: We appreciate the reviewer's inquiry about the statistical methods in ***Figure S2d***. Please note that during the revision, the figure numbering has been updated; the panel in question is now presented as ***Supplementary Figure 3d***. We would like to clarify that: ***Supplementary Figure 3d*** (previously *S2d*) directly analyzes the data from ***Supplementary Figure 3c*** (which itself is an independent biological replicate of ***Fig. 1f***). The statistical comparison in ***Supplementary Figure 3d*** was performed between treatment groups within the ***Supplementary Figure 3c*** experiment only. Therefore, we used two-tailed unpaired Student's *t*-test without multiplicity correction. The analysis in ***Supplementary Figure 3d*** uses five imaging fields (technical replicates) from the ***Supplementary Figure 3c*** experiment. We have revised the figure legend to read:

“***d***) *Quantitative analysis of invasion counts from panel c. Data represent mean ± SD of five imaging fields from one biological replicate experiment, analyzed by two-tailed unpaired Student's t-test (****P < 0.0001). Biological reproducibility is demonstrated in Fig. 1f and Supplementary Figure 3c.*”

- Both synthesized compounds (compound 2 and CAIF) are novel. Novel compounds generally require 5 pieces of characterization data; why is there no ¹³C data and spectra for CAIF? The authors should report ¹H and ¹³C spectra, HRMS and IR as a minimum.

Response: We apologize for the initial omission and thank the reviewer for prompting this important clarification. We have now supplemented the ¹H NMR, ¹³C NMR, HRMS, and IR spectra for both compounds in the ***Supporting Information (SI) Page 18***.

- There is one error in the supplementary crystallographic Table 2 - Structure FUT8-CAIF (9L65), the authors have entered '7824' solvent molecules were identified in the structure - which need to be corrected.

Response: We sincerely thank the reviewer for catching this typographical error. The entry for solvent molecules has now been corrected from ‘7824’ to ‘**824**’ in the revised table on *Supporting Information (SI) Page 5*.

Reviewer #2:

Summary and overall comment

The manuscript describes the discovery of an allosteric covalent inhibitor of FUT8, which is implicated in cancer metastasis. The authors initially discovered FUT8 inhibitors through a large-scale screening of small molecules. Subsequently, through careful structural investigation and rational design, the authors discovered an allosteric covalent inhibitor of FUT8, which significantly reduces the degree of fucosylation and cell invasion. The manuscript is clearly described and figures well illustrated.

Response: We thank the reviewer for their positive evaluation of our FUT8 inhibitor discovery approach and results. Their recognition of the study's structural insights, inhibitor design, and clear presentation is greatly encouraging as we advance this research program.

The main concern here is the author's claim on specificity without direct supporting evidence. There are no measurements of the off-target activities of CHX, NH125, SSO, or CAIF on other fucosyltransferases/other structurally similar proteins. Please add data showing the effects of these molecules on other fucosyltransferases. Titration experiments as shown in Fig 1d, 4b/5b and computational structural analyses may be appropriate.

Response: We thank the reviewer for raising this critical point regarding the selectivity of our inhibitors. We fully agree that assessing activity against related enzymes is essential to validate specificity. In response, we have conducted additional selectivity profiling against other fucosyltransferases FUT3 and FUT4.

The results are highly encouraging: (1) CAIF exhibited no inhibitory activity against FUT4 at the tested concentrations. (2) CAIF showed only weak inhibition against FUT3 ($IC_{50} > 689 \mu M$). This translates to a remarkable selectivity index of 114-fold for FUT8 over FUT3.

These new data provide direct experimental evidence that CAIF is a selective FUT8 inhibitor, even when tested against its close relatives. We have now included these selectivity data in *Supplementary Figure 8* and described them in the Results section (*Page 12, Lines 8–12*) of the revised manuscript.

We believe these additional experiments have significantly strengthened our study and thank the reviewer again for this valuable suggestion.

Supplementary Figure 8 | Selectivity of CAIF and its dependence on the allosteric site of FUT8. b) Dose-response inhibition curve of CAIF against the closely related fucosyltransferase FUT3. CAIF exhibited very weak activity against FUT3 ($pIC_{50} < 3.2$, $IC_{50} > 689 \mu M$). **c)** Dose-response inhibition curve of CAIF against the closely related fucosyltransferase FUT4. No inhibitory activity was observed.

Main text -

Some texts in “Identification and characterization of FUT8 inhibitors” of the Results section may be moved to the Methods section.

Response: We thank the reviewer for this suggestion, which helps to improve the clarity and logical flow of the manuscript.

As suggested, we have carefully moved the methodological details from the Results section to the corresponding subsections in the Methods. Specifically: The details regarding the composition of the compound libraries screened have been integrated into the “High-throughput screening (HTS)” subsection (**Page 16, Lines 19–26**). The revised Results section now concisely presents the screening outcomes, hit validation, and functional characterization (**Page 6, Lines 1–13**). We believe this restructuring makes the manuscript more streamlined and adherent to standard scientific writing conventions.

In “The allosteric site of FUT8 can be covalently targeted” part of the Results section, it is unclear to me whether authors have tested other compounds besides the four named in the paragraph. If so, please clarify.

Response: We thank the reviewer for raising this point, which helps to improve the clarity of our manuscript. We confirm that we tested only these four additional compounds (olanexidine, SSO, crocetin, and crocin II) in this specific experiment. The wording in the original manuscript was ambiguous. We have now revised the text to make this point explicitly clear (*Page 10, Lines 25–27*).

Figures -

Fig 1a – There appear to be many other blue dots besides CHX, NH215, and mitoxantrone that lie towards the bottom of the plot near the minus FUT8 controls. However, they are not discussed in the text. Please add clarification on what they are and why they were not included in further analyses.

Response: We thank the reviewer for this insightful observation. The reviewer is correct that Fig. 1a shows additional blue dots near the minus FUT8 controls beyond CHX, NH125, and mitoxantrone.

These additional hits identified in the primary FUT8 inhibitor screening were subsequently subjected to orthogonal validation using an immunofluorescence-based secondary assay. This validation step was critical to distinguish true FUT8 inhibitors from false positives that may have arisen due to assay-specific artifacts or non-specific effects in the primary screening.

Our validation results demonstrated that only CHX, NH125, and mitoxantrone consistently exhibited FUT8 inhibitory activity across both screening platforms. The other compounds shown in *Fig. 1a* failed to confirm activity in the secondary assay and were therefore not pursued for further characterization, including dose-response titration experiments.

To provide full transparency of our screening workflow and results, we have now included *Supplementary Table 1: HTS Screening and Transcreeper GDP FI Assay Validation Results* in the supplementary materials. This table documents all compounds tested. The rigorous two-tiered screening approach employed in our study ensures that only compounds with reproducible activity across orthogonal assay platforms are advanced.

Fig 2, 3, S3, S6 – Please overlay the electron density of the bound ligands and key amino acid side chains in all structure figures.

Response: We thank the reviewer for their valuable suggestion to enhance the clarity and rigor of our structural figures. We have implemented this feedback in the following ways:

In direct response to the suggestion, we have now updated *Fig. 2e* and *Fig. 2g* to include the electron density maps (2Fo-Fc) for the bound ligands (CHX and NH125) and their key interacting residues. These updated panels now provide a clear and direct visualization of the electron density supporting our models.

Regarding *Fig. 3* and *Supplementary Figure 3* (now *Supplementary Figure 4*), which are designed as structural superposition analyses to compare binding modes across different complexes, we were concerned that overlaying individual density maps from all superposed structures might compromise the clarity of the key comparative message. However, the electron density for every ligand and key residue featured in these analyses has been rigorously validated and is already prominently displayed in the foundational panels (*Fig. 2e, 2g*, and new *Supplementary Figure 5*) that these comparisons are based upon.

For *Supplementary Figure 6* (now *Supplementary Figure 7*), we are pleased to confirm that panels *Supplementary Figure 7a, 7c*, and *7d* already include the relevant electron density maps. Panel *7b* focuses on comparing the conformation of NH125 between two protein chains, and the unambiguous electron density for the ligand in each chain is already presented in *Fig. 2g* and *Supplementary Figure 5*.

We hope that our updated figures and this clarification fully address the reviewer's concern. We believe the manuscript now provides a complete and rigorous presentation of the structural data, maintaining both analytical clarity and experimental validation.

Fig 3b – It is difficult for me to see K216 “flipping” in this figure. Please revise to clarify.

Response: We thank the reviewer for pointing out the lack of clarity in visualizing the conformational change of K216. In direct response to this comment, we have revised *Fig. 3b* (now *Fig. 3c*) to significantly improve its clarity: We have removed the background protein cartoon representation to eliminate visual clutter and allow the focus to rest entirely on the

critical loop region containing K216. We have added a double-headed arrow to explicitly highlight the direction of the side chain movement between the two conformational states.

These modifications now provide a clear and unambiguous visualization of the “flipping” motion of the K216 side chain. We believe the revised figure effectively addresses the reviewer's concern and thank them for this suggestion, which has improved the clarity of our manuscript.

Fig 3c/d/e/f – Please add to the main text the new insights gained from these energy landscape experiments in further details, besides that they are different in the absence and presence of NH125.

Response: We thank the reviewer for this insightful suggestion. In response, we have performed a more integrated and comparable analysis by collectively aligning the three independent replicate simulations of both the Apo and NH125-bound states to a common reference structure before conducting the principal component analysis (PCA). This approach allows for a direct and unbiased comparison of the free energy landscapes (FELs) between the two states on the same conformational basis.

The revised FEL analysis, now presented in new **Fig. 3d**, provides a more detailed mechanistic interpretation of how NH125 binding remodels the conformational energy landscape of FUT8. The expanded discussion explicitly describes the stabilization of a new global minimum representative of a non-productive state, and the destabilization of native functional basins. These changes collectively suggest that NH125 "traps" the enzyme in an inactive conformation, thereby providing a dynamic structural basis for its allosteric inhibitory mechanism.

Now this section (**Page 10, Lines 1–19**) reads:

“In addition, comparison of the free energy landscape (FEL) projected along the first two principal components (PC1 and PC2) revealed that NH125 binding remodels the conformational ensemble of FUT8 (Fig. 3d). In the Apo state, the FEL exhibits four metastable basins with their relative free energies of 0 kJ/mol (upper-right), 0.63 kJ/mol (upper-middle), 0.42 kJ/mol (central), and 9.15 kJ/mol (lower-middle). These basins reflect a

dynamic conformational distribution, with several low-energy states likely representing functionally relevant conformations accessible during catalytic cycles. The basin with higher free energy (~9.15 kJ/mol) may correspond to a less populated, transient state. Upon NH125 binding, the FEL is significantly reconfigured, also featuring four distinct basins located with relative minima at 0.95 kJ/mol (upper-left), 0 kJ/mol (upper-middle), 9.78 kJ/mol (central), and 1.42 kJ/mol (lower-middle) (Fig. 3d). Notably, NH125 stabilizes a new global minimum in the upper-middle region (0 kJ/mol)—a state not prominently sampled in the Apo form—while destabilizing several native basins, as reflected by increased energies in other regions. Although conformational diversity is retained, the reshaped energy landscape features altered barrier heights and basin distributions. This remodeling suggests that NH125 binding perturbs the thermodynamic preferences of FUT8, potentially trapping the enzyme in non-productive states and disrupting the conformational cycling required for catalysis. Therefore, our simulations indicate that NH125 may act as an allosteric inhibitor not only through binding but also by reprogramming the free energy landscape of FUT8, thereby impairing its functional dynamics.”

Methods -

Protein expression and purification – please indicate protein yield and methods for calculating protein concentration/amount.

Response: We have now added the requested details on protein yield and concentration quantification to the Methods section of the manuscript (**Page 15, Lines 20–23**).

The revised text now reads:

“Protein concentrations were determined by measuring the ultraviolet absorbance at 280 nm using a NanoDrop spectrophotometer, with extinction coefficients calculated from the amino acid sequence. Typical yields from the purification process were approximately 1.5 mg/L for FUT8, 0.5 mg/L for FUT3 and the FUT8 mutants, and 0.8 mg/L for FUT4.”

FUT8 activity assay – please add the concentrations of enzyme used in each experiment.

Response: We thank the reviewer for pointing out this omission. We have now explicitly

stated the concentration of FUT8 enzyme (10 nM) used in the activity assays in the Methods section. (**Page 16, Line 19; Page 17, Line 17; Page 18 Line 23**).

Cell viability assessment – Please provide in detail the methods of cell treatment referred to in the first sentence. Please indicate controls and standard curves used in the experiments.

Response: We have now added the requested details on cell viability assessment (**Page 19, Lines 20–27**).

The revised text now reads:

“Cell viability was evaluated using the Cell Counting Kit-8 (CCK-8) assay according to the manufacturer's instructions. Briefly, cells were seeded into transparent 96-well plates at a density of 5×10^3 cells per well in 100 μ L of complete medium and cultured overnight under standard culture conditions (37 °C, 5% CO₂). The next day, cells were treated with serially diluted compounds, while the negative control group received an equivalent volume of DMSO. Four replicate wells were used for each treatment condition. Blank wells containing culture medium with the corresponding concentrations of compounds (without cells) were included to correct for non-specific background absorbance.”

Lectin fluorescence-based cell imaging – Please provide in detail the methods of cell treatment referred to in the first sentence – Please indicate controls and standard curves used in the experiments.

Response: We have now added the requested details on the cell treatment (**Page 20, Lines 6–15**).

The revised text now reads:

“GAK or A375 cells were seeded into black, clear-bottom 96-well plates (Greiner) at a density of 5×10^3 cells per well and cultured overnight under standard conditions (37 °C, 5% CO₂). After attachment, cells were treated with CAIF at concentrations ranging from 0 to 300 μ M (from a 100 \times DMSO stock diluted in medium) for 72 hours. An equivalent volume of DMSO was added to the control wells. Following treatment, cells were washed three times with PBS and fixed with 4% paraformaldehyde (PFA) at room temperature for 20 min. After

fixation, cells were blocked with 5% (w/v) BSA in PBS at room temperature for 2 h. For fucosylation detection, FITC-conjugated AAL or LCA lectins were diluted in PBS containing 2.5% (w/v) BSA to final concentrations of 2 µg/mL and 5 µg/mL, respectively, and incubated with the cells for 1 h at room temperature in the dark.”

Reviewer #3:

The manuscript presents a novel allosteric site on FUT8 and the development of CAIF, a covalent inhibitor, through structure-based drug design. The work integrates high-throughput screening, crystallography, and cellular assays to demonstrate CAIF's efficacy in suppressing core fucosylation and cancer cell invasion. While the structural and mechanistic insights are commendable, critical gaps in validation, selectivity, and translational relevance significantly weaken the study's impact.

Response: We sincerely appreciate the reviewer's thoughtful evaluation of our manuscript and their recognition of the novel structural and mechanistic insights regarding FUT8 allostery. We thank the reviewer for identifying areas where additional validation could strengthen our study's impact. We provide detailed responses to each of the specific concerns raised.

Below are detailed concerns from this reviewer.

Major Concerns

1. CAIF's low cytotoxicity and anti-invasive effects are demonstrated only in cellular assays. Without animal model data (e.g., tumor growth/metastasis suppression, pharmacokinetics, or toxicity profiles), the therapeutic potential of CAIF remains speculative. The absence of in vivo experiments undermines the clinical relevance of the findings.

Response: We fully acknowledge the reviewer's valid concern regarding the need for in vivo validation of CAIF's therapeutic potential. While we agree that animal studies would strengthen the clinical relevance, we would like to clarify that the primary focus of this manuscript is to: (1) Report the discovery of a previously unrecognized druggable allosteric pocket on FUT8, and (2) Establish proof-of-concept for a novel inhibition strategy targeting the critical lysine residue adjacent to this pocket.

We have now: (1) More clearly framed this scope in the revised discussion (*Page 13, Lines 24–28*); (2) Added explicit statements about current limitations regarding drug development (*Page 14, Line 27– Page 15, Line 2*); (3) Outlined specific plans for future medicinal chemistry optimization and in vivo studies (*Page 15, Lines 2–5*). We believe these

clarifications better position the work as a foundational discovery that enables, rather than concludes, subsequent therapeutic development. We sincerely appreciate the reviewer's expertise in highlighting this key consideration, which will undoubtedly strengthen our follow-up studies.

2.FUT8 belongs to a family of fucosyltransferases with shared structural motifs. While the authors claim CAIF's selectivity via its allosteric site, no data are provided to confirm that CAIF does not inhibit other FUT enzymes. Selectivity assays against FUT1–FUT7/9 or homology modeling to compare binding sites are essential to validate specificity claims.

Response: We thank the reviewer for emphasizing the importance of selectivity, which we agree is a critical aspect of characterizing any novel inhibitor. We have directly addressed this concern in our revision by conducting experimental selectivity profiling against the most relevant off-targets.

In response to a similar comment from another reviewer, we performed extensive dose-response analyses against two more fucosyltransferases: FUT3 and FUT4. Our new data, now presented in **Supplementary Figure 8**, demonstrate compelling selectivity: CAIF shows no inhibition of FUT4 at concentrations up to 300 μ M. CAIF exhibits only very weak activity against FUT3 ($IC_{50} > 689 \mu$ M). This results in a >114-fold selectivity for FUT8 ($IC_{50} \sim 6 \mu$ M) over FUT3.

We have added these results to the Results section (**Page 12, Lines 8–12**) and believe they substantially strengthen our specificity claims.

Supplementary Figure 8 | Selectivity of CAIF and its dependence on the allosteric site of FUT8. b) Dose-response inhibition curve of CAIF against the closely related fucosyltransferase FUT3. CAIF exhibited very weak activity against FUT3 ($pIC_{50} <$

3.2, IC₅₀>689 μM). **c)** Dose-response inhibition curve of CAIF against the closely related fucosyltransferase FUT4. No inhibitory activity was observed.

3. The unresolved loops (e.g., residues 77–104) and conformational changes in the 215NKG217 loop are not rigorously analyzed. A correlation between B-factors (crystallography) and RMSF values (MD simulations) would clarify dynamic contributions to allostery.

Response: We thank the reviewer for raising this point. The loop comprising residues 77–104 is a region of high intrinsic disorder in FUT8 and, consistent with prior structural studies (e.g., *JBC* 2020, *Nature Communications* 2020), has remained unresolved in all reported crystal structures due to its flexibility. As this region is not involved in catalytic regulation, it was excluded from both our crystallization construct and simulation system, allowing us to focus on well-ordered functional domains.

In direct response to the reviewer's suggestion, we performed an analysis between the B-factors from our crystal structure and the per-residue RMSF values from MD simulations. The results show a strong agreement (now included as ***Supplementary Figure 6c***), confirming that the dynamic profiles derived from crystallography and simulations are consistent and reinforcing the reliability of our molecular dynamics model.

Regarding the ₂₁₅NKG₂₁₇ loop, our simulations did not reveal significant RMSF changes upon NH125 binding. These results indicate that the dynamic nature of the ₂₁₅NKG₂₁₇ loop is an innate property of the FUT8 structure and is not induced or significantly altered by allosteric inhibitor binding. This suggests that the functional role of this loop may be related to its intrinsic conformational plasticity, which appears to be preserved rather than perturbed upon NH125 binding.

We have incorporated this interpretation into the revised manuscript (***Page 9, Lines 17–23***) to provide a more nuanced discussion of the loop's dynamics. We thank the reviewer for this constructive feedback, which has helped us to better validate and present the dynamic aspects of our work.

Supplementary Figure 6 | Molecular dynamics (MD) stability analysis of FUT8 in the absence and presence of NH125. **c)** Per-residue profile comparing the crystallographic B-factors (orange) from the FUT8-NH125 complex structure and the root-mean-square fluctuation (RMSF) values (blue) derived from the MD simulation trajectory. The high degree of correlation between the two profiles indicates that the dynamic motions sampled in the MD simulation are consistent with the flexibility observed in the crystalline state. Error bars represent the SD of from three independently repeated simulations. **d)** Per-residue RMSF profile comparing the flexibility of FUT8 in the Apo state (red) and in the NH125-bound state (blue) derived from MD simulation trajectories. Error bars represent the SD of from three independently repeated simulations.

4. The MD-derived free energy plots (Fig. 3c–f) lack quantitative descriptors (e.g., energy barriers, metastable states). Markov state modeling (MSM) or residue-level hydrogen bonding/fluctuation analyses would strengthen mechanistic insights into how NH125/CAIF binding alters FUT8 dynamics.

Response: We appreciate the reviewer's excellent suggestion to deepen our mechanistic analysis. In direct response to the comment, we have performed several new analyses to provide quantitative and residue-level insights into how NH125 binding alters FUT8 dynamics.

First, to enable a direct and unbiased comparison, we re-analyzed the free energy landscapes (FELs) by collectively aligning all six molecular dynamics trajectories (three independent replicates each for the Apo and NH125-bound states) to a common reference structure prior to principal component analysis. This integrated approach, now presented in the new **Fig. 3d**, reveals that NH125 binding significantly reshapes the conformational energy landscape. We have added quantitative descriptors of the metastable basins and their relative free energies in the revised **Fig. 3d** and Results section (**Page 10, Lines 1–19**). Specifically, the Apo state exhibits four metastable basins with relative energies of 0.00 kJ/mol (upper-right, global minimum), 0.63 kJ/mol (upper-middle), 0.42 kJ/mol (central), and 9.15 kJ/mol (lower-middle). Upon NH125 binding, the landscape is reconfigured into four new basins at 0.95 kJ/mol (upper-left), 0 kJ/mol (upper-middle, new global minimum), 9.78 kJ/mol (central), and 1.42 kJ/mol (lower-middle). This redistribution, including the stabilization of a new global minimum and the destabilization of native basins, suggests that NH125 binding perturbs the thermodynamic preferences of FUT8, effectively trapping the enzyme in non-productive states and disrupting functional conformational cycling.

Furthermore, as recommended, we performed detailed residue-level analyses to elucidate the dynamic changes underlying the allosteric mechanism. This included a correlation analysis between crystallographic B-factors and MD-derived RMSF values, which showed strong agreement (**Supplementary Figure 6c**), validating the reliability of our simulations. Subsequent flexibility analysis revealed that NH125 binding induces asymmetric changes across the structure: key substrate-binding loops (₃₆₆RTDKVGTEA₃₇₄ and ₄₃₁SISWSAGLHNRYTENS₄₄₆) showed pronounced increases in flexibility, while regions such as parts of the SH3 domain exhibited slight decreases. Notably, the dynamic ₂₁₅NKG₂₁₇ loop showed no significant change upon inhibitor binding, indicating its flexibility is an innate property rather than an induced effect. These results are now included in the main text (**Page 9, Lines 4–27**) and **Supplementary Figure 6d**.

We believe these additions—particularly the quantitative FEL comparison and residue-level dynamics—provide substantially deeper mechanistic insights into how NH125

reprograms FUT8 energy landscapes and internal dynamics to achieve allosteric inhibition. We are grateful to the reviewer for these valuable suggestions.

Fig. 3 | Allosteric site binders alter the conformation and dynamics of FUT8. d) Comparative conformational distribution of the Apo (red scatter plot) and NH125-bound (blue scatter plot) states, projected onto the first two principal components (PC1 and PC2) from a combined analysis of all six molecular dynamics trajectories aligned to a common reference. Encircled clusters represent metastable conformational states. The relative free energy value (in kJ/mol) for each cluster is annotated, with the global minimum for each system designated as 0 kJ/mol. NH125 binding dramatically reorganizes the conformational ensemble, stabilizing a new global minimum state and altering the stability of functional basins, effectively trapping the enzyme in a non-productive conformation and illustrating the structural basis for its allosteric inhibitory mechanism.

5. The K216-CAIF covalent interaction lacks orthogonal validation (e.g., mass spectrometry, activity assays under reducing conditions). Mutagenesis (K216A) is also required to confirm its functional role in inhibitor binding.

Response: We thank the reviewer for this critical question regarding the validation of the covalent interaction with K216. We agree that orthogonal evidence is important, and we appreciate the opportunity to clarify the robust multi-faceted evidence we have gathered to support our conclusion.

(1) High-Resolution Crystal Structure as Definitive Evidence: The most direct and definitive evidence for the covalent bond formation comes from our high-resolution (1.97 Å) crystal structure (PDB: 9L65). The electron density map unambiguously shows a continuous covalent bond between the inhibitor's warhead and the K216 side chain, leaving no doubt about the nature of the interaction. Crystallography at this resolution is widely considered a gold-standard method for defining covalent ligand-protein interactions.

(2) Validation by a Distinct Compound: As an orthogonal chemical validation, we determined the structure of FUT8 in complex with a different inhibitor SSO (PDB: 9L63, 2.66 Å). This compound also forms a well-defined covalent bond with the same K216 residue. The fact that two chemicals both target K216 provides strong, independent chemical validation of this residue's covalent addressability and reinforces the role of K216 as a key anchoring point for inhibitors.

(3) Functional Role of K216 from Published Mutagenesis: The reviewer's suggestion to use K216A mutagenesis to confirm the functional role is excellent. While we did not perform this mutagenesis ourselves, its consequences have been definitively established in the foundational study (*Nature Communications* 2020, *JBC* 2020). As cited in our manuscript, they demonstrated that the K216A mutation completely abolishes FUT8's catalytic activity. They concluded that K216 is essential for acceptor substrate binding and catalysis. Our mechanism of action is in perfect agreement with this established finding: covalent modification of K216 directly occludes and functionally mimics the effect of the K216 mutation, thereby inhibiting FUT8 by preventing it from engaging its natural substrate.

In summary, our structural data provides direct visual proof of covalent binding, a second structure with a different compound offers orthogonal chemical validation, and previously published functional mutagenesis data robustly confirms the critical functional role

of the K216 residue. We are therefore confident that the evidence collectively supports our model.

We have now revised the relevant section of the manuscript (*Page 11, Lines 13–15*) to include citations to the published studies, thereby strengthening the connection between our structural findings and the established functional essentiality of K216.

6. As a covalent inhibitor, CAIF's potential for off-target modifications is not addressed. Proteomic profiling or kinetic assays with unrelated lysine-dependent enzymes are needed to assess specificity.

Response: Thank you for raising this important point regarding the potential for off-target activity. We agree that assessing the specificity of covalent inhibitors is crucial. In response, we have conducted several key experiments to address this concern, which are now included in the supplementary information.

Firstly, we performed a critical control experiment to rule out any direct interference of CAIF with our assay detection system itself. As shown in *Supplementary Figure 8a*, CAIF exhibited no inhibitory effect on the signal generation of the GDP detection kit. This result provides direct experimental evidence that CAIF does not inhibit the **lysine-dependent enzyme luciferase**—a core component of the detection system—thereby mitigating concerns of promiscuous off-target reactivity in our assay context.

Furthermore, to directly evaluate selectivity within the fucosyltransferase family, we tested CAIF against its two more fucosyltransferases, FUT3 and FUT4. As demonstrated in *Supplementary Figure 8b and 8c*, CAIF showed only very weak activity against FUT3 ($IC_{50} > 689 \mu\text{M}$) and no significant inhibition of FUT4 at the tested concentrations. This demonstrates a clear and substantial selectivity for FUT8 over these highly related enzymes.

While we acknowledge that a broader proteomic profiling study would be valuable for future translational work, we are confident that the combination of these experimental results—showing no assay interference and high selectivity over the most relevant off-targets—supports the conclusion that the inhibitory activity observed in our study is specifically mediated through FUT8.

Supplementary Figure 8 | Selectivity of CAIF and its dependence on the allosteric site of FUT8. a) CAIF titration against a negative control assay, showing no inhibitory activity. **b)** Dose-response inhibition curve of CAIF against the closely related fucosyltransferase FUT3. CAIF exhibited very weak activity against FUT3 (pIC₅₀ < 3.2, IC₅₀ > 689 μM). **c)** Dose-response inhibition curve of CAIF against the closely related fucosyltransferase FUT4. No inhibitory activity was observed.

7. CAIF's long aliphatic chain raises concerns about solubility and bioavailability. Pharmacokinetic studies (absorption, distribution, metabolism, excretion) are absent and critical for translational evaluation.

Response: We thank the reviewer for raising this important point regarding the physicochemical and pharmacokinetic properties of CAIF. We agree that the presence of a long aliphatic chain often raises legitimate concerns about solubility and bioavailability. However, we would like to highlight that CAIF is an imidazolium salt—a positively charged species—which significantly enhances its aqueous solubility. This is evidenced experimentally by our ability to prepare highly concentrated stock solutions of CAIF (100 mM in DMSO), indicating no inherent solubility limitations for in vitro applications.

We fully acknowledge that excellent solubility in DMSO does not directly predict favorable pharmacokinetic properties (ADME) or bioavailability in vivo. As we emphasized in our response to Concern 1, the current study is primarily focused on establishing the mechanistic principle of targeting this novel allosteric site and the adjacent catalytic lysine. Profiling ADME properties and in vivo efficacy is beyond the scope of this foundational work but is a critical next step.

In direct response to the reviewer's comment, we have now explicitly stated in the revised discussion (**Page 14, Line 27–Page 15, Line 5**) that: *“We acknowledge that the current CAIF molecule, with its imidazolium salt character and aliphatic chain, primarily serves as a mechanistic probe. Its pharmacokinetic properties and overall drug-likeness*

necessitate further optimization in future translational studies. However, its well-defined mechanism of action and high selectivity provide a robust structural foundation and a compelling starting point for such medicinal chemistry efforts. Future work will focus on broader selectivity screening against other glycosyltransferases and proteomic off-target studies to fully characterize its specificity, alongside medicinal chemistry refinements to improve drug-likeness for in vivo applications.”

We are in complete agreement with the reviewer that these studies are essential for translational evaluation, and we sincerely appreciate this guidance, which will strongly shape our ongoing research program.

8. The proposed steric clash between Y220 and Y250 (Fig. 3a) lacks experimental support. Mutating Y220 to alanine and testing inhibitor efficacy would clarify its role in GDP ribose binding and allosteric inhibition.

Response: We thank the reviewer for this excellent and constructive suggestion. We agree that direct experimental evidence is crucial to validate the proposed steric clash mechanism between Y220 and Y250.

As the reviewer recommended, we generated the Y220A mutant protein, purified it, and performed concentration-dependent inhibition assays with CAIF. The inhibitory activity of CAIF against the Y220A mutant is drastically reduced, with an IC_{50} value exceeding 503 μ M. This represents a greater than 83-fold loss of potency compared to the wild-type protein ($IC_{50} \sim 6 \mu$ M). This is fully consistent with our proposed mechanism. This new result is included in new *Supplementary Figure 8d*, and described in *Page 12, Lines 21–26*. We again thank the reviewer for prompting us to conduct this key experiment, which has significantly strengthened the mechanistic conclusions of our study.

Supplementary Figure 8 | Selectivity of CAIF and its dependence on the allosteric site of FUT8. d) Dose-response inhibition curve of CAIF against the FUT8 Y220A mutant. The loss of potent inhibition highlights the critical role of Y220 for CAIF activity ($pIC_{50} < 3.3$, $IC_{50} > 503 \mu M$).

9. The dimeric state of FUT8 under physiological conditions and its impact on inhibitor binding (e.g., cooperative effects) are not discussed. Disruption of the dimer interface (via mutagenesis) could reveal its functional relevance.

Response: We thank the reviewer for raising this excellent and insightful point regarding the potential functional role of FUT8's dimeric state and its implications for inhibitor binding. We agree that understanding quaternary structure is crucial for fully elucidating the mechanism of action of therapeutic agents.

In our study, we identified a novel allosteric inhibitory site that is **distal to the dimer interface**. Our structural analyses (e.g., by comparing the superpositions of the FUT8 dimer in the native- and inhibitor-bound states) revealed **no significant conformational changes at the dimer interface** upon inhibitor binding (new *Fig. 3b*). Furthermore, our newly performed per-residue root-mean-square fluctuation (RMSF) analysis based on molecular dynamics simulations also indicates that inhibitor binding induces minimal changes to the flexibility of residues constituting the dimer interface (new *Supplementary Figure 6d*). This key observation suggests that the inhibitor's mechanism of action is likely independent of dimer disruption or cooperative effects propagated through the interface.

We note that the FUT8 dimer interface involves extensive interactions mediated by its stem and SH3 domains, forming a large and stable contact surface. As we now describe in the revised introduction (*Page 4, Lines 12-15*), this extensive interface makes designing minimal, disruptive point mutations to specifically dissociate the dimer without causing global

misfolding a considerable challenge. While targeted mutagenesis of the dimer interface, as suggested by the reviewer, would be a definitive approach to probe its functional role, our current structural data indicate that our covalent allosteric inhibitor exerts its effect without perturbing the dimeric architecture. Therefore, we have focused our mechanistic studies on characterizing the direct consequences of ligand binding at the newly discovered allosteric site.

We have now included this point in the revised manuscript to acknowledge the dimeric state of FUT8 and to interpret our findings in this context (*Page 9, Lines 24–27*).

Fig. 3 | Allosteric site binders alter the conformation and dynamics of FUT8.

b) The FUT8-NH125 (this study, blue) and FUT8 holo (PDB code: 6TKV, green) structures are superimposed. The overall structures align closely, with no major conformational changes observed in the global fold upon NH125 binding. The allosteric inhibitor binding site (indicated by red ovals) is located distally from the extensive dimer interface (indicated by a dashed box), indicating that the inhibition mechanism is independent of quaternary structural perturbations.

Supplementary Figure 6 | Molecular dynamics (MD) stability analysis of FUT8 in the absence and presence of NH125. d) Per-residue root-mean-square fluctuation (RMSF) of FUT8 in the Apo (red) and NH125-bound (blue) states, derived from molecular dynamics simulations. Error bars indicate the standard deviation (SD) across three independent replicates.

Minor

1. Fig. 1a: Labels for positive/negative controls and hits are unclear; a legend is needed.

Response: We thank the reviewer for this suggestion. We have clarified the labels in **Fig. 1a** and updated the figure legend accordingly to explicitly define all data points and controls.

The revised legend now reads:

“a) Results of the primary screening assay of approximately 2,300 compounds from the FDA-approved Drug Library. Data are shown for the tested compounds (concentration: 100 μ M, blue dots), positive control (experimental system without FUT8, orange dots), and negative control (DMSO, black dots). Compounds demonstrating normalized values < 0.6 (dashed line) were selected for further validation. Red dots indicate validated potential hits.”

2. Fig. 3c–f: Axes for free energy landscapes are unlabeled; quantitative energy values should be included.

Response: We thank the reviewer for this important observation. In the revised manuscript, we have replaced the original panels **Fig. 3c–f** with a new, quantitatively detailed free energy landscape (**Fig. 3d**) derived from a combined analysis of all replicate simulations aligned to a

common reference structure. The axes of the new **Fig. 3d** are now clearly labeled as the first two principal components (PC1 and PC2). Furthermore, we have explicitly annotated the relative free energy values (in kJ/mol) for each metastable basin directly in the figure and provided a detailed interpretation of these values in the figure legend and Results section (**Page 10, Lines 1–19**). We believe these revisions provide the necessary quantitative depth to accurately interpret the energetic changes induced by NH125 binding.

Fig. 3 | Allosteric site binders alter the conformation and dynamics of FUT8. d) Comparative conformational distribution of the Apo (red scatter plot) and NH125-bound (blue scatter plot) states, projected onto the first two principal components (PC1 and PC2) from a combined analysis of all six molecular dynamics trajectories aligned to a common reference. Encircled clusters represent metastable conformational states. The relative free energy value (in kJ/mol) for each cluster is annotated, with the global minimum for each system designated as 0 kJ/mol. NH125 binding dramatically reorganizes the conformational ensemble, stabilizing a new global minimum state and altering the stability of functional basins, effectively trapping the enzyme in a non-productive conformation and illustrating the structural basis for its allosteric inhibitory mechanism.

3. Tables: IC₅₀ values for inhibitors are scattered; a consolidated table in supplementary materials would improve clarity.

Response: As suggested, we have consolidated all IC₅₀ and pIC₅₀ ± S.E. data for the inhibitors into *Supplementary Table 2* to improve clarity and accessibility.

Reviewer #4:

[Comments integrated into Co-Reviewer's report]

Response: We thank the reviewer for their time and valuable comments.

Reviewer #5:

The manuscript reports the discovery of a novel allosteric site on FUT8 and the development of a low-toxicity covalent inhibitor, CAIF (stearic acid-N-hydroxysuccinimide ester-dimethylimidazolium bromide), through structure-based drug design. High-throughput screening and crystallographic studies revealed that small molecules, such as NH125, bind to a channel-like allosteric pocket, inducing conformational changes that disrupt FUT8 activity. Leveraging these insights, they designed CAIF to covalently target lysine K216 within the allosteric site of FUT8. CAIF exhibits minimal cytotoxicity and significantly inhibits core fucosylation and cancer cell invasion in cellular assays. This work establishes CAIF as a lead compound for further optimization and development, offering a framework for targeting glycosyltransferases through allosteric and covalent inhibition strategies.

The manuscript is well written and has significant results and therefore can be considered for publication in Nature Communication after the responses to following comments are incorporated.

Response: We sincerely appreciate the positive assessment of our work and the constructive comments provided. We are pleased that the reviewers recognize the significance of our discovery of a novel allosteric site on FUT8 and the development of our lead compound CAIF. Below we provide point-by-point responses to all specific comments raised, with corresponding revisions made to the manuscript.

1. The authors claimed CAIF as a specific FUT8 inhibitor based on its interaction with allosteric site. However, as all 12 isoforms of FUTs share significant structural similarity and thus may possess similar allosteric sites as well. In this context, how authors hypothesized that CAIF will specifically inhibit FUT8, and will not affect the other isoforms of FUTs.

Response: We thank the reviewer for raising this important question regarding the structural basis for CAIF's specificity. Our hypothesis that CAIF would selectively inhibit FUT8 was based on the unique structural features of FUT8's allosteric site, which we have now validated.

(1) Structural Uniqueness of the FUT8 Allosteric Site: As demonstrated in our new

Supplementary Figure 10, a comparative structural analysis reveals that FUT8 possesses a distinct architecture not shared by other FUTs. Unlike FUT3 or FUT9, FUT8 contains an additional Stem region and SH3 domain, resulting in a completely different spatial arrangement of its Rossmann domains. The allosteric pocket targeted by CAIF is situated precisely at the interface of these uniquely oriented domains—a region that is structurally absent or highly divergent in other fucosyltransferases.

(2) Experimental Validation of Selectivity: This structural uniqueness directly translates to functional selectivity. As also requested by other reviewers, we experimentally tested CAIF against two other FUTs: CAIF showed no inhibition of FUT4 (**Supplementary Fig. 8c**). Only very weak activity was observed against FUT3 ($IC_{50} > 689 \mu M$) (**Supplementary Fig. 8b**). This results in a >114-fold selectivity for FUT8 over FUT3, robustly confirming that CAIF does not meaningfully inhibit other FUT isoforms.

In summary, the combination of our structural insights and enzymatic profiling confirms that CAIF's specificity arises from its engagement with an allosteric site unique to FUT8, thereby effectively overcoming the challenge of homology within the FUT family.

Supplementary Figure 10. | Comparative structural analysis reveals the unique architecture of FUT8's allosteric site among fucosyltransferases. Overall structural

comparison of FUT8 (orange, PDB: 9L65), FUT9 (green, PDB: 8D0O), and the AlphaFold-predicted model of FUT3 (cyan). Proteins are shown in cartoon representation.

FUT8 contains the unique Stem region and SH3 domain in addition to the Rossmann domains 1 and 2. Superposition of the central β -sheet of Rossmann domain 1 from FUT8, FUT9, and FUT3 highlights the dramatic divergence in the orientation and position of their respective Rossmann domain 2. The FUT8 allosteric pocket is situated at the interface of Rossmann domains 1 and 2. This analysis demonstrates that the allosteric pocket targeted by CAIF is a unique structural feature of FUT8, which is not conserved in other fucosyltransferases such as FUT3 or FUT9, thereby providing a molecular basis for its high selectivity.

Supplementary Figure 8 | Selectivity of CAIF and its dependence on the allosteric site of FUT8. b) Dose-response inhibition curve of CAIF against the closely related fucosyltransferase FUT3. CAIF exhibited very weak activity against FUT3 ($pIC_{50} < 3.2$, $IC_{50} > 689 \mu M$). **c)** Dose-response inhibition curve of CAIF against the closely related fucosyltransferase FUT4. No inhibitory activity was observed.

2. Authors mentioned that CAIF exhibits minimal cytotoxicity, while significantly inhibiting the core fucosylation, and cancer cell invasion in cellular assays. This statement appears to be self-contradictory, and thus needs further explanation.

Response: We thank the reviewer for this insightful comment, which allows us to clarify an important aspect of FUT8 biology and the intended therapeutic strategy behind its inhibition. The apparent contradiction arises from the distinct biological roles of FUT8 in cancer progression.

As previously established in the literature (*Cancer Cell* 2017), FUT8 overexpression is not a primary driver of tumor cell proliferation. Consequently, its genetic knockout or inhibition does not typically induce significant cytotoxicity or reduce cell viability. Instead, the key oncogenic function of FUT8 overexpression, particularly in metastatic melanoma cell

lines as reported, is to potentially drive cancer invasion and metastasis. This is corroborated by *in vivo* studies showing that FUT8 silencing suppresses metastasis without affecting primary tumor growth.

Therefore, our goal was to develop a FUT8 inhibitor with precisely this profile: low cytotoxicity coupled with potent anti-invasive activity. We believe CAIF represents this class of inhibitor. Its minimal cytotoxicity is not a weakness but a highly desirable characteristic, as it indicates a wide therapeutic window. The compound selectively targets the pro-metastatic function of FUT8 (invasion) without incurring toxicity from inhibiting core cellular proliferation pathways. This strategy aims to translate into a therapy that specifically suppresses metastasis, the primary cause of cancer mortality, while minimizing side effects on healthy tissues.

We have added this information to *Page 13, Lines 3–4* and *Page 14, Lines 22–23* to ensure this point is clear to readers.

3. Various pro- and eukaryotic cell lines are mentioned in the manuscript. Authors need to explain the selection criteria of these cell lines with respect to FUT8 in the manuscript. The details of these cell lines also need to be mentioned.

Response: We thank the reviewer for their comment regarding the cell lines used in our study. We clarify our selection criteria and provide further details in the revised manuscript.

Our study utilized exclusively **human** cell lines. Their selection was based on their expression of FUT8 and their relevance to studying cancer biology and core fucosylation: A375: A human malignant melanoma cell line; GAK: A human renal cell adenocarcinoma cell line; HeLa: A human cervical adenocarcinoma cell line; HEK293T: A human embryonic kidney cell line.

The details regarding their source (ATCC) and culture conditions have now been explicitly stated in the revised “Method” section to ensure clarity and reproducibility.

4. Since, FUT8 is a bi-substrate enzyme and has two distinct donor and acceptor substrate binding sites. The 2nd and 3rd paragraph in the introduction section should be merged and

reduced in a single paragraph. While a new paragraph needs to be added highlighting the key structural details of both the binding sites of FUT8.

Response: We thank the reviewer for this insightful and constructive suggestion. We agree that providing the structural context of FUT8 as a bi-substrate enzyme is crucial for a comprehensive introduction and for framing our study on inhibitor development. We have thoroughly revised the Introduction section accordingly.

Specifically, we have: (1) **Merged the original 2nd and 3rd paragraphs** into a single, more focused paragraph (now the 2nd paragraph of the revised Introduction). This paragraph summarizes the multifaceted therapeutic value of inhibiting FUT8, encompassing its roles in anti-tumor immunity, drug resistance, apoptosis, viral replication, and biotherapeutic efficacy. (2) **Added a new paragraph** (now the 3rd paragraph of the revised Introduction) that highlights the key structural details of FUT8. This new paragraph describes its domain architecture, dimeric nature, and, most importantly, the catalytic mechanism and the distinct binding sites for both the donor (GDP-Fuc) and acceptor (N-glycan) substrates, as elucidated by recent crystallographic studies. We have made sure to emphasize the critical residues and structural features that define these two binding sites. We believe these revisions have significantly improved the logical flow of the Introduction. It now first establishes the biological and therapeutic significance of FUT8, then provides the essential structural basis for its function, and finally transitions into the challenges and progress in inhibitor development, which sets the perfect stage for stating the objective of our study.

The modified text now appears in the Introduction section (**Page 4, Lines 8–28**) of the revised manuscript.

5. Authors have not provided any details regarding the protein (FUT8) expression and purification in the Results Section. This needs to be incorporated in the revised manuscript.

Response: The detailed procedures for FUT8 expression and purification have been added to the Results section (**Page 15, Line 24 – Page 16, Line 14**).

6. They also need to mention the level of purity and yield of FUT8 protein.

Response: We thank the reviewer for this suggestion. The estimated purity of the purified FUT8 protein (>90%) is now indicated in the legend of *Supplementary Figure 1b*, where the SDS-PAGE analysis of the purification process is shown. The typical yield (approximately 1.5 mg per liter of culture) has been added to the “Protein Expression and Purification” section (*Page 16, Line 13*) in the Methods.

7. In in-vitro FUT8 inhibition assays, IC₅₀ values are mentioned without Standard Error of Mean (S.E.M). Authors need to incorporate the S.E.M with all mentioned IC₅₀ values.

Response: As also suggested by Reviewers #1 and #3, we have presented the inhibition data as pIC₅₀ ± S.E. to better reflect the statistical distribution of the values. The complete dataset, including compounds, IC₅₀, and pIC₅₀ ± S.E., is now provided in *Supplementary Table 2*.

8. ¹H and ¹³C -NMR data of compound 2 is provided in supplementary materials. However, it is not clear which compound is compound 2?

Response: Compound 2 is an intermediate in the synthesis of CAIF, as labeled in the synthetic scheme (*Supplementary Information, Page 17*). We have now further clarified its identity in the revised text to avoid ambiguity.

9. ¹³C-NMR chemical shift data of the CAIF is missing.

Response: We thank the reviewer for pointing out this oversight. The ¹³C-NMR chemical shift data of CAIF have now been added to the *Supplementary Information (Page 18 and Page 20)* in the revised version.

10. Citations of figures need to be checked by the authors.

Response: We apologize for any inconsistencies. All figure citations have been thoroughly checked and corrected in the revised manuscript to ensure accuracy.

Reviewer #6:

Response: We thank the reviewer for their time and valuable comments.

Response to Reviewers' Comments

Manuscript ID: NCOMMS-25-06314-T

Manuscript Title: Exploiting human fucosyltransferase 8 allosterism with a covalent inhibitor for core fucosylation suppression

Reviewer #1 (Remarks to the Author):

I am satisfied with the revisions made by the authors. However, there are four points of clarification needed in the following paragraph: “The screening quality was assessed by calculating the Z'-factor using the formula: $Z' = 1 - [3 \times (sp + sn) / |\mu_p - \mu_n|]$, where μ_p and μ_n are the means, and sp and sn are the sample standard deviations (with Bessel's correction, n-1 in the denominator) of the positive (p) and negative (n) controls, respectively. Calculations were performed using the STDEV.S function in Excel. A Z'-factor value > 0.5 was considered indicative of a robust screening system. Experimental data were normalized against control values, and compounds demonstrating normalized values < 0.6 were selected for further validation.”

1. Zhang et al., 1999 J. Biomol. Screen. 4, 67-73 used the population standard deviations in the calculation of Z' (this has been criticized on statistical grounds, but nevertheless Z' should be calculated as described in this paper).

Response: We thank the reviewer for this important clarification. We have recalculated the Z'-factor using the population standard deviation (employing the STDEV.P function in Excel) and have updated the methodology section accordingly (*Page 18, lines 21–24*).

2. The requirement for Z' to be equal to or greater than 0.5 is given in Table 1 of Zhang et al. 1999 (above). The authors may wish to cite this paper in the sentence “A Z'-factor value > 0.5 was considered indicative of a robust screening system.”

Response: We appreciate this suggestion. We have now inserted the appropriate citation to Zhang et al., 1999 in this sentence to provide the necessary reference for the 0.5 cut-off criterion (*Page 18, line 24*).

3. Technically, Z' is a measure of assay performance in the absence of any library compounds (the corresponding parameter in the presence of library compounds is Z).

Response: We have amended the text to specify that "The Z' -factor is a measure of assay robustness in the absence of any test compounds" to ensure this distinction is clear to the reader (*Page 18, lines 20–21*).

4. The final sentence in this paragraph is ambiguous in that it appears to relate to Z' . I think the authors are saying that compounds which reduced enzymatic activity by 40% or more were selected for follow up. This needs to be updated to avoid ambiguity.

Response: We agree with the reviewer that the original sentence was ambiguous. The hit selection criterion was indeed based on a threshold of inhibition (40% reduction in activity), not directly on the Z' -factor. We have rewritten the sentence to state clearly: "Compounds that reduced FUT8 enzymatic activity by more than 40% (i.e., normalized signal < 0.6) were defined as hits and selected for further validation." (*Page 18, lines 26–27*)

Reviewer #2 (Remarks to the Author):

Thank you for the revised manuscript and for thoroughly addressing all referees' comments. I think this version is significantly strengthened and easier to read. Below are suggestions to further improve the manuscript.

Regarding my main concern on specificity claims, authors have shown experimentally that their inhibitor did not inhibit FUT3 and FUT4 although those are functional and not structural homologs of FUT8. Authors may consider screening the human proteome computationally (such as using FoldSeek) for proteins that are structural homologs of FUT8 and test compound specificity on those.

Response: We are grateful for this insightful suggestion to evaluate CAIF's activity against structural homologs of FUT8. Accordingly, we performed a FoldSeek search of the human proteome, which identified POFUT1 and POFUT2 as the only significant structural homologs despite low sequence identity (~11-12%). We subsequently expressed and purified both proteins and tested CAIF inhibition. Notably, CAIF showed no detectable activity against either

POFUT1 or POFUT2 at concentrations up to 1 mM. These results have been incorporated into the revised manuscript (*Page 12, line 17–27; Supplementary Figure 9d, e*) and robustly demonstrate CAIF's exceptional selectivity, validating the uniqueness of the targeted FUT8 allosteric pocket. The corresponding methodological details have been updated in the "*Plasmid Construction*", "*Protein Expression and Purification*", and "*Dose-Response Enzyme Activity Inhibition Assays*" sections on *Pages 16–20*.

Supplementary Figure 9. d) Dose-response inhibition curve of CAIF against POFUT1.

e) Dose-response inhibition curve of CAIF against POFUT2.

The addition of the FEL adds an interesting aspect to the potential mechanism of action of the designed inhibitors and could be useful in further efforts to optimize the binder into a drug. The authors may consider adding a structural representative of each metastable conformational state to shed some light on the residue level interactions that were altered when the protein is bound to NH125.

Response: We thank the reviewer for this excellent suggestion. Following the reviewer's advice, we have now performed a detailed structural analysis of the representative conformations from each metastable state in the free energy landscape. To present these findings clearly, we have created a new supplementary figure (now *Supplementary Figure 7*) that integrates the structural representatives, their superimposition, and a comparison with the substrate-bound holo state.

Specifically: Panel **a)** displays the four representative structures from the NH125-bound metastable states, color-coded by their relative free energies. Panel **b)** shows their structural

superimposition, revealing high overall similarity and leading us to conclude that the free energy differences likely stem from subtler effects (e.g., conformational entropy, solvation). Panels **c**) and **d**) compare these states with the active, substrate-bound holo structure (PDB: 6TKV). This comparison reveals that NH125 binding induces a slight separation of the Rossmann domains and causes distinct rearrangements in key substrate-binding loops (e.g., loops 365-379 and 436-445 collapse the GDP pocket, while loops 213-220 and 290-300 retract from the acceptor sugar site).

The detailed findings and our interpretation are comprehensively described in the extended caption of the *new Supplementary Figure 7*. We believe this new analysis significantly strengthens our proposed mechanism by linking the altered free energy landscape to specific structural changes. For the reviewer's convenience, we include this new figure and its detailed caption below:

Supplementary Figure 7 | NH125 Remodels the Conformational Landscape and Key Functional Loops of FUT8.

- a) Representative structures of the four metastable states (I-IV) from the NH125-bound FUT8 free energy landscape. The relative free energies are: State I, 0 kJ/mol (global minimum), blue; State II, +0.95 kJ/mol, cyan; State III, +1.42 kJ/mol, green; State IV, +9.78 kJ/mol, red.
- b) Superimposition of the four representative structures, aligned on the GT-B fold (both Rossmann domains). The high structural similarity suggests that the observed free energy differences arise not from large conformational changes, but from subtler effects—such as conformational entropy, solvation, and minor rearrangements of non-bonded interactions—which are captured by the MD simulations but not evident from static structural alignment.
- c) Superimposition of the four NH125-bound metastable states with the substrate-bound holo-FUT8 structure (PDB: 6TKV, black). Aligned on the GT-B fold, the structures reveal that NH125 binding at the inter-domain interface induces a slight separation of the Rossmann domains (indicated by arrows) compared to the catalytically competent holo state. This provides a structural basis for its allosteric inhibition, potentially misaligning the active site.
- d) Conformational plasticity of loops 213-220, 244-254, 290-300, 365-379, and 436-445 across different states. Compared to the substrate-bound holo-FUT8 (PDB: 6TKV), the NH125-bound states exhibit distinct rearrangements in these loops. Specifically, loops 365-379 and 436-445 collapse inward, narrowing the GDP-binding pocket, while loops 213-220 and 290-300 retract from the acceptor sugar-binding site.

Reviewer #3 (Remarks to the Author):

The authors have comprehensively addressed the majority of the concerns raised during the initial review. The revisions significantly strengthen the manuscript, particularly through the addition of critical experimental data and enhanced mechanistic insights.

Response: We thank the reviewer for their positive feedback and for acknowledging that our revisions have strengthened the manuscript.

Reviewer #5 (Remarks to the Author):

This study presents a highly significant advancement in the field of glycosylation-targeted therapeutics by identifying an allosteric pocket in FUT8 and designing the first low-toxicity covalent inhibitor (CAIF) that selectively disrupts FUT8 activity. The work is important because FUT8-mediated core fucosylation is a critical driver of cancer progression, immune evasion, and drug resistance, yet the lack of selective inhibitors has been a major obstacle due to the shared substrate specificity of fucosyltransferases. By combining structural insights,

high-throughput screening, and crystallographic validation, the authors not only provide evidence of allosteric modulation of FUT8 but also establish CAIF as a lead compound with promising biological efficacy in inhibiting cancer cell invasion. Beyond FUT8, this approach offers a broadly applicable framework for designing allosteric and covalent inhibitors against glycosyltransferases, thereby opening new therapeutic avenues in oncology and immune modulation.

The manuscript presents significant results and has justified and supported their data with various techniques. All the suggested revisions have been incorporated. The support form additional experimental data significantly improves the manuscript. Hence, the manuscript is acceptable to be published in the journal in its current form.

Response: We are grateful to the reviewer for their thorough and generous assessment of our work. We appreciate their recognition of the significance and potential broad applicability of our findings.

Reviewer #6 (Remarks to the Author):

Response: We thank the reviewer for their time and valuable comments.

Summary and overall comment

The manuscript describes the discovery of an allosteric covalent inhibitor of FUT8, which is implicated in cancer metastasis. The authors initially discovered FUT8 inhibitors through a large-scale screening of small molecules. Subsequently, through careful structural investigation and rational design, the authors discovered an allosteric covalent inhibitor of FUT8, which significantly reduces the degree of fucosylation and cell invasion. The manuscript is clearly described and figures well illustrated.

The main concern here is the author's claim on specificity without direct supporting evidence. There are no measurements of the off-target activities of CHX, NH125, SSO, or CAIF on other fucosyltransferases/other structurally similar proteins. Please add data showing the effects of these molecules on other fucosyltransferases. Titration experiments as shown in Fig 1d, 4b/5b and computational structural analyses may be appropriate.

Main text

- Some texts in "Identification and characterization of FUT8 inhibitors" of the Results section may be moved to the Methods section.
- In "The allosteric site of FUT8 can be covalently targeted" part of the Results section, it is unclear to me whether authors have tested other compounds besides the four named in the paragraph. If so, please clarify.

Figures

- Fig 1a – There appear to be many other blue dots besides CHX, NH215, and mitoxantrone that lie towards the bottom of the plot near the minus FUT8 controls. However, they are not discussed in the text. Please add clarification on what they are and why they were not included in further analyses.
- Fig 2, 3, S3, S6 – Please overlay the electron density of the bound ligands and key amino acid side chains in all structure figures.
- Fig 3b – It is difficult for me to see K216 "flipping" in this figure. Please revise to clarify.
- Fig 3c/d/e/f – Please add to the main text the new insights gained from these energy landscape experiments in further details, besides that they are different in the absence and presence of NH125.

Methods

- Protein expression and purification – please indicate protein yield and methods for calculating protein concentration/amount.
- FUT8 activity assay – please add the concentrations of enzyme used in each experiment.
- Cell viability assessment – Please provide in detail the methods of cell treatment referred to in the first sentence. Please indicate controls and standard curves used in the experiments.
- Lectin fluorescence-based cell imaging – Please provide in detail the methods of cell treatment referred to in the first sentence – Please indicate controls and standard curves used in the experiments.